



# Effects of the prewhitening method, the time granularity and the time segmentation on the Mann-Kendall trend detection and the associated Sen's slope

Martine Collaud Coen[1], Elisabeth Andrews[2,3], Alessandro Bigi[4], Gonzague Romanens[1], Giovanni Martucci[1] and Laurent Vuilleumier[1].

[1] Federal Office of Meteorology and Climatology, MeteoSwiss, Payerne, Switzerland
[2] Cooperative Institute for Research in Environmental Sciences, University of Colorado, Boulder, CO, USA
[3] NOAA/Earth Systems Research Laboratory Boulder, CO, USA
[4] Università di Modena e Reggio Emilia, Department of Engineering "Enzo Ferrari", Modena, Italy

Correspondence: Martine Collaud Coen (martine.collaudcoen@meteoswiss.ch)

## Abstract

The most widely used non-parametric method for trend analysis is the Mann-Kendall test associated with the Sen's slope. The Mann-Kendall test requires serially uncorrelated time series, whereas most of the atmospheric processes exhibit positive autocorrelation. Several prewhitening methods have been designed to overcome the presence of lag-1 autocorrelation. These include a prewhitening, a detrending and/or a correction for the detrended slope and the original variance of the time series. The choice of which prewhitening method and temporal segmentation to apply has consequences for the statistical significance, the value of the slope and of the confidence limits. Here, the effects of various prewhitening methods are analyzed for seven time series comprising in-situ aerosol measurements (scattering coefficient, absorption coefficient, number concentration and aerosol optical depth), Raman Lidar water vapor mixing ratio and the tropopause and zero degree levels measured by radio-sounding. These time series are characterized by a broad variety of distributions, ranges and lag-1 autocorrelation values and vary in length between 10 and 60 years. A common way to work around the autocorrelation problem is to decrease it by averaging the data over longer time intervals than in the original time series. Thus, the second focus of this study is evaluation of the effect of time granularity on long-term trend analysis. Finally, a new algorithm involving three prewhitening methods is proposed in order to maximize the power of the test, to minimize the amount of erroneous detected trends in the absence of a real trend and to ensure the best slope estimate for the considered length of the time series.

Keywords: Seasonal Mann-Kendall test, Theil-Sen's slope, prewhiten, detrend, autocorrelation

## 1. Introduction

To estimate climate changes and to validate climatic models, long-term time series associated with statistically adapted trend analysis tools are necessary. The basic requirements needed to apply specific statistical tools are usually well described, but end-users often do not systematically test if the properties of their time series fulfill these requirements. An inappropriate usage of the statistical tools may lead to



misleading conclusions. It may also happen that a time series does not meet the complete criteria of any of the statistical tools. In that case, the statistical tool can be adapted or the use of different methods with complementary strengths and weaknesses should be applied.

The time series properties that can cause misuse of statistical tools for trend analysis primarily concern the statistical distribution, the autocorrelation, missing data or periods without measurements, the presence of seasonality, irregular sampling, the presence of negatives and the rules applied in the case of data below-detection limits. A large number of trend analysis tools such as the whole family of least mean square and generalized least squares methods are parametric methods and, consequently, require normally distributed residues. Unfortunately, many atmospheric measurements, which strongly depart from the normal distribution, do not meet this requirement so that non-parametric methods have to be used. Non-parametric techniques are commonly based on rank and assume continuous monotonic increasing or decreasing trends. The Mann-Kendall (MK) test associated with the Sen's slope is the most widely applied non-parametric trend analysis method in atmospheric and hydrologic research (Gilbert, 1987; Sirois, 1998). While it has no requirement on data distribution, it must be applied on serially independent and identically distributed variables. The second condition of homogeneity of distribution is not met if a seasonality is present, but it can be solved by using the seasonal Mann-Kendall test developed by Hirsch et al. (1982). The first condition of independence is not met if the data are autocorrelated, which is often the case where atmospheric variables are controlled by autocorrelated physical or chemical processes. To analyze properly autocorrelated and not normally distributed errors, two different strategies are usually applied as described below.

The first strategy tends to decrease the autocorrelation by aggregating time series into monthly, seasonally, yearly data or even in longer periods. However, coarser time granularities (e.g., due to longer averaging periods) do not ensure autocorrelation is removed. Moreover, the aggregation implies a decrease of the information density in the time series, such as the diurnal or seasonal cycles, the variance of the data and to some extent the data distribution. The aggregation conditions (length of the time unit, making the time unit consistent with the observed seasonality, starting phase of the time series and the averaging method) may influence the trend results (de Jong and de Bruin, 2012; Maurya, 2013) in what is called the Modifiable Temporal Unit Problem (MTUP).

The second strategy focuses on the development of algorithms to reduce the impact of the autocorrelation artifacts on the statistical significance of the MK test and on the Sen's slope. Two kinds of algorithms are usually used: (i) the prewhitening of the data to remove the autocorrelation and (ii) inflation of the variance of the trend test statistic to take into account the number of independent measurements instead of the number of data points (the autocorrelation reduces the number of degrees of freedom in tests).

In this study, the effects of various prewhitening methods on the MK statistical significance and on the slope are analyzed for time series of in situ aerosol properties, aerosol optical depth, temperature levels (tropopause and zero degree levels) and remote sensing water vapor mixing ratio. This study also analyses the effect of the time granularity on the MK statistical significance, on the strength of the slope and on the confidence limits of various atmospheric compounds for the atmospheric time series listed above. Additionally, a new methodology combining three prewhitening methods is proposed in order to handle correctly the autocorrelation without decreasing the power of the test while still computing the correct slope value.



## 2.  The Mann-Kendall methodology (prewhitening methods)

The MK-test for trends is a non-parametric method based on rank. The calculated S statistic is normally distributed for a number of observation N>10 and the significance of the trends is tested by comparing the standardized test statistic $Z=S/[var(S)]^{0.5}$ with the standard normal variate at the desired significance level. For N≤10, an exact S distribution has to be applied (see e.g., Gilbert, 1987). Hirsch et al. (1982) extend the Mann-Kendall test to take seasonality in the data into account as well as multiple observations for each season. A global or annual trend can be considered only if the seasonal trends are homogeneous at the desired confidence level (Gilbert, 1987). Confidence limits (CL) are defined as the 100(1-p) percentiles of the standard normal distribution of all the pairwise slopes computed during the Sen's slope estimator, where p is the chosen confidence limit.

### 2.1  The problem of the autocorrelation in the MK-test

The MK-test determines the validity of the null hypothesis $H_0$ of the absence of a trend against the alternative hypothesis $H_1$ of the existence of a monotonic continuous trend. While no assumptions are needed about the data distribution (i.e., the definition of a non-parametric test), the MK-test does require that the data are serially independent, namely the absence of autocorrelation in the time series. Statistical tests are prone to two types of error. The first is an incorrect rejection of the null hypothesis $H_0$ that is called "type 1 error". This error is related to a too high statistical significance leading to false positive cases. The second is an incorrect acceptance of the null hypothesis $H_0$ that is called "type 2 error". This error can be understood as the power of the test being too low leading to false negative cases.

The adverse effect of the positive autocorrelation in time series on the number of type 1 errors was suggested by Tiao et al. (1990) and Hamed and Rao (1998) and later simulated (Kulkarni & von Storch, 1995, Zwang and Zwiers, 2004, Blain, 2013, Wang et al., 2015, Hardison et al., 2019). All these studies clearly showed that positive autocorrelation in time series largely increases the number of type 1 errors, whereas prewhitening procedures increased the number of type 2 errors. Larger lag-1 autocorrelation ($ak_1$) leads to higher percentage of type 1 errors and to larger bias in the Sen's slope. Zwang and Zwiers (2004) also show that the occurrence of both types of error largely depends on the length of the time series, with longer periods leading to a strong reduction of errors and to a lower bias in the trend slope estimation.

A popular solution to get rid of the autocorrelation problem in the MK-test is to aggregate the time series in order to decrease $ak_1$. While the use of coarse time granularity effectively decreases the autocorrelation, the suppression of autocorrelation is not guaranteed, even in monthly or yearly aggregations. Moreover, aggregation greatly decreases the number of observations N and can potentially affect the MK-test errors, the slope biases and the CL.

Two kinds of statistical procedures were developed to correct the MK-test for autocorrelation in the data. The variance correction approaches (Hamed and Rao, 1998; Yue and Wang, 2004; Hamed 2009; Blain, 2013) consider inflating the variance of the S statistic so that the number of independent observations instead of the total number of observations is taken into account. These approaches appear not able to



preserve the significance level and the power of the MK-test in the case of correlated time series with a trend (Yue et al., 2002; Blain, 2013). The prewhitening approaches consider removing the lag-1 autoregressive (AR(1)) process in the time series prior to applying the MK-test. Several algorithms with various strengths and defaults have been published and are described in the next section. Since negative

autocorrelations are rare in atmospheric processes, only positive autocorrelations are taken into account in this study. Several studies have shown that the prewhitening methods are also applicable in case of negative serial correlations but with dissimilar consequences (Rivard and Vigneault, 200, Yue and Wang, 2002,  Bayazit et al., 2004).

## 2.2  The prewhitening methods

This section describes all the prewhitening methods known to the authors. The advantages and disadvantages of each method are summarized in Table 1. It has to be noted that, for all the methods proposed, the prewhitening can be applied only if $ak_1$ is statistically significant (ss) following a normal

distribution at the two-sided 95% confidence interval. The first implemented prewhitening method (hereafter called PW) simply removes the lag-1 autocorrelation $ak_1^{data}$ from the original data X at the time t:

$$X_t^{PW} = X_t - ak_1^{data} X_{t-1} \tag{1}$$

This PW method results in a low amount of type 1 errors, but it reduces the power of the test due to an

over-/underestimation of $ak_1^{data}$ in the case of a positive/negative trend. A further procedure called trend-free prewhitening (TFPW) consists of removing the autocorrelation on detrended data. Yue et al. (2002) published the most commonly used method that consists of: i) estimating the Sen's slope $\beta^{data}$ on the original data; ii) removing the trend to obtain a detrended time series $A^{detr}$ (eq. 2); iii) removing the lag-1 autocorrelation $ak_1^{detr}$ on $A^{detr}$ to  generate a detrended prewhitened time series $A^{detr-prew}$ (eq. 3); and  iv)

adding the trend back in to generate the processed time series to evaluate (i.e., $X_t^{TFPW-Y}$) (eq. 4):

$$A_t^{detr} = X_t - \beta^{data} t \tag{2}$$

$$A_t^{detr-prew} = A_t^{detr} - ak_1^{detr} A_{t-1}^{detr} \tag{3}$$

$$X_t^{TFPW-Y} = A_t^{detr-prew} + \beta^{data} t \tag{4}$$

This approach is called trend-free prewhitening (TFPW-Y) and restores the power of the test, albeit at the expense of an increase of type 1 errors. Wang and Swail (2001) propose an iterative TFPW method that consists of: i) removing $ak_1^{data}$ from the original time series and correcting the prewhitened data for the modified mean (eq. 5); ii) estimating the Sen's slope $\beta^{prew}$ on the prewhitened data $A_{cor,t}^{prew}$; iii) removing the trend ($\beta^{prew}$) estimated on the PW data from the original data to obtain a prewhitened detrended time

series $A_{cor,t}^{detr}$(eq. 6); and iv) applying iteratively i-iii until the $ak_1$ and slope differences become smaller than a proposed threshold of 0.05 (eq. 7).

$$A_{cor,t}^{prew} = X_t^{PW-cor} = (X_t - ak_1^{data} X_{t-1})/(1 - ak_1^{data}) \tag{5}$$





$$A_{cor,t}^{detr} = (X_t - \beta^{\ prew} t) \tag{6}$$

$$A_{cor,t}^{detr-prew} = (A_{cor,t}^{detr} - ak_1^{detr-prew} A_{cor,t-1}^{detr})/(1 - ak_1^{detr-prew}) \tag{7}$$

$$X_t^{TFPW-WS} = A_{cor,t}^{detr-prew} \tag{8}$$

After n iterations until $ak_1^{detr-prew,n-1} - ak_1^{detr-prew,n} < 0.05 \ and \ \beta^{prew,n-1} - \beta^{prew,n} < 0.05$

Wang and Swail's (2001) TFPW method (TFPW-WS) restores the low number of type 1 errors without decreasing the power of the test (Zhang and Zwiers, 2004). The factor $(1-ak_1^{detr-prew})^{-1}$ is needed to ensure that the prewhitened time series possesses the same trend as the original time series. The PW-cor method refers to the preliminary step of the first iteration in the TFPW-WS method and consequently corrects the prewhitened data by the same factor. To the knowledge of the authors, this PW-cor method is not referenced in the literature but is a potential method tested in this study.

Finally, Wang et al. (2015) proposed a further approach in order to correct TFPW-Y for both the elevated variance of slope estimators and for the decreased slope caused by the prewhitening. Practically, the variance of $A^{detr-prew}$ (i.e., $\sigma_A^2$) is restored to the variance of X (i.e., $\sigma_X^2$) to generate the $A_{VC}^{detr-prew}$ time series:

$$A_{VC,t}^{detr-prew} = A_t^{detr-prew} * \frac{\sigma_X^2}{\sigma_A^2} \tag{9}$$

The slope estimator $\beta^{data}$ is decreased in the case of positive autocorrelation by the square root of the variance inflation factor (VIF) to obtain the corrected slope $\beta_{VC}^{detr}$ (eq. 11). Matalas and Sankarasubramanian (2003) provided a simple way to compute the limiting values of VIF for a sufficiently large sample size and a first order autocorrelation:

$$VIF \approx (1 + ak_1^{detr})/(1 - ak_1^{detr}) \tag{10}$$

So that

$$\beta_{VC}^{detr} = \beta^{\ data}/\sqrt{(1 + ak_1^{detr})/(1 - ak_1^{detr})} \tag{11}$$

and

$$X_t^{VCTFPW} = A_{VC,t}^{detr-prew} + \beta_{VC}^{detr} t \tag{12}$$

Statistical simulations by Wang (2015) showed that this new variance corrected prewhitening method (VCTFPW) leads to more accurate slope estimators, preserves to some extent the power of the test, but only mitigates the type 1 errors.



### 2.3 A new algorithm involving three prewhitening methods


As described in sect. 2.2 and Table 1, each of the presented prewhitening methods has a primary advantage: the low type 1 error for PW, the high-test power for TFPW-Y and the unbiased slope estimate for VCTFPW. TFPW-WS has both a low type 1 error and a high test power, but requires more computing time due to the iteration process. We propose a new algorithm, described in Fig. 1, which combines the

advantages of each prewhitening method:

- The $ak_1^{data}$ of the original time series is calculated. If it is not ss, the MK test is applied on the original time series. If $ak_1^{data}$ is ss, PW, TFPW-Y and VCTFPW are applied in order to obtain three prewhitened time series that are thereafter named after the specific prewhitening method for purposes of clarity.

- The MK-test that defines the statistical significance is applied on the PW and TFPW-Y data. If both tests are ss or not ss, the trend is considered as ss or not ss, respectively. If TFPW-Y is ss but not PW, the trend is considered as a false positive due to the too high type 1 errors of TFPW-Y and the trend has to be considered as not ss. If PW is ss but TFPW-Y is not, then the trend is considered as a false negative due to the lower test power of PW and the trend has to be considered as ss.

- The Sen's slope is then computed on the VCTFPW data in order to have an unbiased slope estimate.

### 3. Experimental

In order to have a broader view of the effects of the various PW methods, several very different time series (Table 2) were used: three surface in-situ aerosol properties (absorption coefficient, scattering coefficient and number concentration) measured at Bondville (BND), a remote, rural station in Illinois, USA; the aerosol optical depth (AOD) measured at Payerne (PAY) on the Swiss plateau; the tropopause and the zero-degree levels measured by radio-sounding launched at PAY; and the water vapor mixing ratio

at 1015 m measured by remote sensing at PAY. The shortest time series (AOD and water vapor mixing ratio) cover only 10 years (y) of measurements while the longest time series cover 60 y. The three in-situ aerosol properties are Johnson-distributed and diverge strongly from a normal distribution. The other time series exhibit distributions that also diverge from a normal distribution but to a lower extent so that some of them have residuals of a least mean square fit, which are normally distributed. The values of

some of the time series span over several orders of magnitude and the scattering and absorption coefficients time series contains negative values due to detection limit issues in very clean conditions. The zero-degree level time series also includes negative altitudes, since it is interpolated to altitudes lower than sea level in the case of negative ground temperature at PAY (S. Bader et al., 2019)). All the data have high $ak_1^{data}$ at the daily time granularity and exhibit clear seasonal cycles with maxima in summer.

Trend analyses were applied on several periods. For all the data sets, a 10-year period is considered first and then further possible multi-decadal periods up to 60 y for the radio-sounding time series. For the in-situ aerosol properties, tests with 4 to 9 y periods are also computed in order to illustrate the problems of trend analysis on very short time series. The number of data points in the time series (N) depends on the length of the period and on the time granularity. The choice of temporal segmentation to address

seasonality for the seasonal MK-tests can also affect N and was evaluated by segmenting the time series into months and meteorological seasons for time granularities up to one month. The MK-test was also applied on the complete time series without considering seasonality (no temporal segmentation) for



comparison purposes, even though, properly, seasonal MK-tests must be used when seasonal cycles are present.

To assess the statistical significance, the two-tailed p-values are computed. For a more comprehensive presentation of the results, the statistical significance is presented here as 1 minus p-value so that the ss at a 95% confidence level is effectively given by ss=0.95. If not further specified, the ss of the trend and of $ak_1{}^{data}$ is given at the 95% confidence level, whereas CL are given at the 90% confidence level. The slopes (in percent) are normalized by the median of the data. Periods of at least 10 y and trends on these periods
are further called decadal periods and decadal trends.

## 4.  Results and discussion

As explained in the methodology section (Sect. 2), the trend results (e.g., the ss, the slopes and the CL)
depend on a number of factors, the most important factors being the prewhitening method, the number of data points in the time series and the autocorrelation. The choice of the prewhitening method clearly affects the ss, the slope and the CL. Analysis choices such as the time granularity, the length of the analyzed period and the temporal segmentation to address seasonality affect $ak_1{}^{data}$, N and the variance of the time series. There is a pronounced interdependency among these variables involving critical choices
in the presentation of the results. Some general plots are first presented to provide insights into the primary results for some of the time series. They are followed by a more detailed analysis of the effects of the prewhitening method, the time granularity, the temporal segmentation, the length of the data series and the number of data points in the time series.

MK trend results (Fig. 2) of the aerosol number concentration, the aerosol absorption coefficient, the
tropopause level and the AOD are plotted as a function of the time granularity for the MK-test and for all the prewhitening methods. The results are shown for no temporal segmentation (circle) and for two different temporal segmentations to address seasonality (meteorological seasons (square) and months (triangle)). The three aerosol properties exhibit decreasing trends while the results of the tropopause level time series indicate a positive trend. The negative aerosol slopes are related to the decreasing aerosol
load in Western Europe and North America (Collaud Coen et al., 2020, Yoon et al., 2016). The increasing tropopause level trend is related to global warming (Xian and Homeyer, 2019). The results of the trends will not be further described and discussed, since this study is only focused on the methodology of the trend analysis.

The common features for all the time series considered here are:

- As described in Wang et al. (2015), the absolute value of the VCTFPW slopes lies between the TFPW and the PW slope values. The absolute value of the PW slopes is always smaller than the TFPW slope values.
- The MK, TFPW-Y, TFPW-WS and PW-cor methods result in similar slopes.
- Large time aggregations usually lead to not ss $ak_1{}^{data}$ and, consequently, prewhitening methods
do not need to be applied. The $ak_1{}^{data}$ of all prewhitening methods is not ss for three-month aggregations of the tropopause level and AOD datasets and for one-year aggregation of the


aerosol absorption coefficient and AOD. The $ak_1^{data}$ of the aerosol number concentration remains ss until the one-year aggregation.

- CL are smaller for finer time granularities in the presence of ss $ak_1^{data}$.
- CL of MK, PW and TFPW-Y, which remove the lag-1 autocorrelation without compensation for the mean values and the variances, are smaller than for VCTFPW, PW-cor and TFPW-WS. PW-cor and TFPW-WS have the highest CL.
- The ss often decreases for coarser time granularities occasionally leading to not ss trends for some of the prewhitening methods. PW, TFPW-WS and VCTFPW methods become not ss at finer time granularities than TFPW-Y and MK due to their lower number of false positives.
- The discrepancies between prewhitening methods are larger than the discrepancies that occur when different temporal segmentations (months or meteorological seasons) are applied.

Apart from these common results, there are features that depend on the time series, such as the effects of the applied temporal segmentation to address seasonality, the similarity of MK slopes with TFPW slopes, and the time granularity leading to not ss $ak_1^{data}$. For example, the number of data points in the AOD time series (about 65 per year) induces higher CL for time granularities finer than the measurement frequency (about 10 days).

## 4.1  Effects of the prewhitening methods

As predicted theoretically, the ss depends on the prewhitening method, with higher ss for the MK and TFPW-Y methods that are related to higher type 1 errors (false positives), while PW and VCTFPW have a lower ss and a lower test power. This is verified on the individual time series, e.g., for the aerosol number concentration results presented in Fig. 3a. The yearly trend was computed for all periods (from 5y to 24y) at all considered time granularities (1 day to 1 month for the meteorological season temporal segmentation), leading to 40 trends. The results show:

- The MK-test ss without prewhitening has a median of 1, with the ss for the upper quartile and upper whisker also equal to 1 and thus within the 95% confidence level so that only 5 trends out of 40 evaluated (i.e., 12.5%) are not ss.
- The TFPW-Y ss has a median slightly lower than 1 and only 3 trends (7.5%) outside the 95% confidence level.
- The TFPW-WS ss has a median of 0.996 which is lower than MK and TFPW-Y. The lower quartile for TFPW-WS, is 0.89, which is outside the 95% confidence level and indicates that 32.5% of the trends are not ss.
- The results of both PW and PW-cor are similar to the TFPW-WS with median ss of 0.995, a lower quartile of 0.84 and 32.5% of the trends are not ss.
- The VCTFPW ss has the lowest median (0.98), first quartile (0.83) and lower whisker (0.63) leading to 37.5% of trends being not ss.

Similar results are found for all time series, but with less difference amongst the methods when the trends are obviously present or absent and more differences for weak trends.

According to Monte-Carlo simulations presented in the literature (e.g. Yue et al., 2002, Wang et al., 2015, Hardison et al., 2019), TFPW-Y leads to a high number of false positives. Since this study deals with





measured data, the rate of false positives is defined as trends that are ss with TFPW-Y but not ss with PW, since the latter is the method with the lowest rate of type 1 error. Figure 3b shows that the number of

false positives depends, as expected, on the strength of the slope and on $ak_1^{data}$. Weaker trends (smaller slopes in percent) are usually associated with lower ss and consequently lead to a larger number of false positives. The impact of the PW and TFPW-Y depends largely on $ak_1^{data}$ absolute values, i.e., higher $ak_1^{data}$ leads to stronger modification of the original time series with lower means (e.g., the mean of $X_t^{PW}$ is less than the mean of $X_t$) and reduced variances for positive $ak_1^{data}$. The highest $ak_1^{data}$ values (between 0.85

and 0.9) found in the time series studied lead to 60% to 100% false positives while $ak_1^{data}$ values between 0.8 and 0.85 lead to at least 40%  false positives.

To obtain a better view of the weakness of each MK-test, the percentage of false positives and false negatives are reported in Table 3 for all the datasets. PW is used as the reference for false positives because it is the prewhitening method with the lowest type 1 error, while TFPW-Y is the reference for

false negatives because it is the most powerful test. For the decadal trends, MK, TFPW-Y and VCTFPW have 33-49% of false positives. This suggests that half of the trends determined using VCTFPW are false positives. TFPW-WS has less than 2% of false positives whereas PW-cor has similar false positives as PW. While PW, PW-cor and TFPW-WS have a low percentage of false negatives, false negatives make up ~5% of the trends for MK and up to one third for VCTFPW. For the trends on short periods, the lower amounts

of type 1 and 2 errors for MK and TFPW-Y are due to the overestimation of the slopes with these tests (see section 4.4) leading to more robust trends and enhanced ss. The unbiased estimate of the VCTFPW slope produces similar amounts of errors for the short-term trends as for the decadal trends. While the choice of PW as reference to compute the number of type 1 errors is obvious (Zhang and Zwiers, 2004, Yue et al., 2002, Blain, 2013, Wang et al. 2015), MK could also be considered as an alternative reference

for the power of the test instead of TFPW-Y. If MK is the power of test reference, then the TFPW-Y percentage of false negatives is 9.4% for the decadal trends and 3.5% for the short-term trends. MK and TFPW-Y then each result in 3-10% of false negatives, however the false negatives are for different cases for the two tests. For the time series considered in this study, the following conclusions can be made: 1) PW performs very well with an almost vanishingly small (≤0.3%) number of false negatives and the ss of

PW-cor is similar to that for PW; 2) TFPW-WS has a very low amount of both type 1 and 2 errors; 3) VCTFPW has a very high type 1 and 2 errors and should consequently not be used to determine the ss; and 4) it is not possible to determine whether MK or TFPW-Y is the most powerful method.

The effects of the prewhitening method on the slope (Fig. 2 and 4) also follow the theoretically deduced assumptions:

● The slope of the trend is always enhanced by the positive $ak_1^{data}$, which adds a multiple of the t-1 value to the t value (e.g., Eqn 1 and 3). By removing the autocorrelation, PW leads to a strong decrease in the absolute value of the slope that becomes smaller than the MK slope. The $CL_{PW}$ are also somewhat decreased (Fig. 5) due to the decreased mean and variance of the prewhitened time series, relative to the original dataset.

● Due to the detrending procedure, the absolute values of the TFPW-Y slope are larger than the PW slopes and similar to the MK slope values (Fig. 2), even if a tendency to have larger TFPW-Y than MK slopes are observed (Fig. 4b). The $CL_{TFPW-Y}$ are similar to the $CL_{PW}$ because the variance and mean are similar for both the PW and TFPW-Y prewhitened time series.

● Due to the corrected slope and variance, the absolute values of the VCTFPW slopes are much

smaller than the TFPW-Y slopes but larger than the PW slopes.





These theoretical assumptions are validated in all cases with the ss trends analyzed in this study. The water vapor mixing ratio and the zero degree level both have a very high autocorrelation (about 0.9 at one-day time granularity). In such cases, the removal of the autocorrelation can lead to not ss trends and the absolute values of the VCTFPW slope are not always larger than PW slope values.

The slope difference among the methods depends directly on $ak_1{}^{data}$. A more nuanced estimate of the slope dependence is shown in Fig. 4 where the differences among the prewhitening methods are plotted. As already mentioned, the VCTFPW method largely mitigates the slope overestimate of the TFPW-Y method at large $ak_1{}^{data}$ so that the increase of the slope absolute value for increasing $ak_1{}^{data}$ does not exceed a factor of two (100% difference in Fig. 4a). The difference between VCTFPW and
TFPW-Y slopes can reach 200-1000% for the largest $ak_1{}^{data}$. The overestimation of the slope by TFPW-Y is much larger than the underestimation by PW if VCTFPW is taken as a reference for slope estimation. TFPW-Y slopes tend to be larger than MK slopes (Fig. 4b), with larger differences at high $ak_1{}^{data}$ leads. Finally, the slope difference between MK and both TFPW-WS and PW-cor does not depend on $ak_1{}^{data}$ and the TFPW-WS and PW-cor slopes are usually nearly identical as suggested by their similar
relationship to the MK slope (Fig. 4c-d).

The effects of the prewhitening method on CL (Fig. 5) are explained by their modification of the mean and the variance of the data. Removing the lag-1 autocorrelation increases the variance, but decreases the mean. The correcting factor of $(1-ak_1)^{-1}$ used in the TFPW-WS and PW-cor methods restores the mean (eq.
5), whereas the VCPWTF method restores the initial variance (eq. 9). All increases of the variance make the CL interval wider, whereas the decrease of the mean decreases the CL interval. $CL_{TFPW-Y}$ and $CL_{PW}$ are the narrowest due to lower mean and variance values while $CL_{TFPW-WS}$ and $CL_{PW-cor}$ are the widest due to larger variance induced by the prewhitening and a mean identical to the original data. $CL_{VCTFPW}$ are intermediate with a variance similar to the original data but a lower mean.


## 4.2   Effects of the time granularity

Averaging is often used to decrease $ak_1{}^{data}$ in the time series. To investigate this, the $ak_1{}^{data}$ values are plotted as a function of the time granularity for the last 10 y of all the time series (Fig. 6a). The decrease
of $ak_1{}^{data}$ with aggregation does not have a large impact until granularity is coarser than one-month. For one-month time granularity and less, aggregation leads to an $ak_1{}^{data}$ difference smaller than 0.2 in 5 of the time series. Three-month and one-year aggregation involve a sharper reduction of $ak_1{}^{data}$. Additionally $ak_1{}^{data}$ for one-year aggregation is, for most of the time series, no longer ss and, sometimes, even negative. The decrease in $ak_1{}^{data}$ is not continuous with time granularity, with $ak_1{}^{data}$ often larger for 10 days or one
month than for 3 days aggregation. These local minima can be explained by a competitive effect between the $ak_1{}^{data}$ decrease and a reduction of the measurement variance. The spread of the slopes of the aerosol number concentration for the one-year aggregation on Fig. 2c shows that the yearly data still have a ss $ak_1{}^{data}$ for the longest periods of 20 and 24 years (see similar cases in Fig. 2). For shorter periods (5 to 9 years), the $ak_1{}^{data}$ decreases rapidly for averaging longer than 10 days and even becomes negative for
yearly averages.

TFPW-Y and TFPW-WS remove the autocorrelation computed from the detrended data. Fig. 6b and 6c show the difference in $ak_1$ between the original and the detrended time series as a function of the time granularity. The $ak_1{}^{detr}$ continuously increases with aggregation whereas $ak_1{}^{detr-prew,n}$ sometimes decreases (e.g., for one-month or three-months aggregations for scattering coefficient and number concentration,





respectively). While the differences in $ak_1$ from the original time series are larger for TFPW-WS than for TFPW-Y, they remain relatively small and exceed 0.05 only in few cases.

Figure 7 presents the effect of the time granularity on ss of the trends for the zero degree level data set for different periods (identified by colors) and various prewhitening methods (identified by symbols). MK and PW-cor are not included since their ss values are nearly identical to the TFPW-Y and PW ss values,
respectively. As expected, TFPW-Y exhibits the highest ss, followed by TFPW-WS, while PW and VCTFPW exhibit the lowest ss. The ss always decreases at coarser time granularities for all prewhitening methods until $ak_1^{data}$ becomes not ss, usually at an average of 3 months. This decrease in ss is larger for the PW, TFPW-WS and VCTFPW than for TFPW-Y. For robust trends analyzed (e.g., the period of 40 y in Figure 7), the trend remains ss at the 95% or 90% confidence level for the finest time granularity (3 days for PW and
TFPW-WS and 1 month for TFPW-Y ), but this is often not the case for weak trends.

When $ak_1^{data}$ is not ss at high time granularity, the prewhitening methods can no longer be applied and the ss is similar for all methods. Without prewhitening, the ss is inversely proportional to the variance reduction caused by the aggregation. For TFPW-Y, the removal of the prewhitening due to not ss $ak_1^{data}$ at three months aggregation corresponds however to a decrease of the ss of the trend. The $ak_1^{detr-prew,n}$ of
the 40 y period is ss for the one-year time granularity as can be seen by the TFPW-WS ss that is different than the ss of the other prewhitening methods (Fig. 7), leading to lower ss than without prewhitening. The increase of the ss with the period length is also obvious, with smaller differences between TFPW-Y and PW for longer periods. The longest period (40 y) and the finest time granularities (1-3 days) lead to no false positives for TFPW-Y, which is not the case for shorter periods or coarser time granularities.

The effect of the time granularity on the slope strongly correlates with the $ak_1$ time granularity dependence. A decrease of the autocorrelation with aggregation induces a reduction of the prewhitening effects on the slopes leading to a decrease in the differences between slopes (see Figs. 2 and 4).

The loss of ss with coarser time granularities is even more pronounced when evaluated for each month or meteorological season (Fig. 8). This is due to the lower N per season (1/4 for meteorological season
and 1/12 for months). Similarly, the decrease in the difference in slopes due to aggregation and the reduction of the prewhitening effects is more pronounced when temporal segmentation is applied due to the reduction of the number of data points in each temporal segment.

Fig. 8 clearly shows that the coarsest time granularities enhance the variability for the different temporal segmentation choices. For example, the interval between the minimum and maximum slopes is 2.3 larger
for the monthly average than for the daily average for the scattering coefficient temporally segmented into  12 months (Fig. 8a) and 3.7 times larger for the absorption coefficient with meteorological seasons (Fig. 8b), respectively. In some cases, the sign of the slope changes with the time granularity when the trends are not ss. As already observed in Fig. 2, the CL also increase with time granularity due to the decrease in N. The effects of the time granularity on the ss, the slope and the CL are more pronounced for
a monthly than for meteorological seasons temporal segmentation due N being three times lower for the months than it is for the seasons.




### 4.3   Effects of temporal segmentation to address seasonality

The division of the year into temporal segments is a necessary condition of the MK-test if the data exhibit
a clear seasonality. Statistically, it is important to have equivalent segments with similar lengths to obtain
similar N per segment. The time series presented in this study are all dependent on phenomena related
to the temperature (e.g., atmospheric circulation, boundary layer height, source changes, etc.), and thus
change with the meteorological seasons. The seasonality of time series primarily affected by other
meteorological phenomena (e.g., the Asian monsoon, which is better characterized by dry and humid
seasons, rather than the standard 4 meteorological seasons) have to be carefully studied in order to
choose both the appropriate temporal segmentation and the appropriate time granularity. For example,
a time granularity that does not respect the seasonal variation of a time series can lead to erratic results
(de Jong and de Bruin, 2012).

The effects of the chosen temporal segmentation to address seasonality are presented here for the
VCTFPW slope and CL, but they are similar for the other methods as well. The effect of including temporal
segmentation on the ss of the yearly trend is rather small with a difference of only 2-3% in the number of
ss trends (not shown). The division into four meteorological seasons always results in the largest number
of ss trends, while the division into 12 months is less powerful for short periods due to the low number of
points for each month (N ≤ 10) for a 10 y period.  The application of no temporal segmentation, which
does not met the MK-test requirements in the presence of a seasonality, is less powerful for decadal
trends. No systematic effects due to the choice of temporal segmentation on the slope were found.
Different temporal segmentation choices lead, most of the time, to comparable slopes. The effect of the
prewhitening method is always much more pronounced than the effect of the choice of temporal
segmentation.

Figure 9 presents the CL intervals normalized by the trend slope as a function of the time granularity for
the aerosol scattering coefficient without temporal segmentation (blue) or divided into monthly (green)
or meteorological seasons (red) for several periods between 5 y and 24 y. Due to the decrease of N, finer
temporal segments induce an increase of the CL. In the case presented in Fig. 9, monthly segments have
CL intervals four times larger than when seasonality is not considered and 2 times larger than
meteorological seasons for the longest periods. It should be recalled, however, that not considering
seasonality for time granularity finer than one-year is not allowed due to the observed seasonal variation
in the aerosol scattering coefficient time series.

In the case of a seasonal MK-test, yearly trend results can be considered only if the trends are
homogeneous among the temporal segments (see Sect. 2.1). The division of the time series into four
meteorological seasons leads to more homogeneous trends (three times and 25 times for decadal and
short periods, respectively) at the 90% confidence level than the division into 12 months (Table 4). Thus,
if meteorological seasons correspond to the observed temporal cycle of the studied time series then those
seasons should be the preferred temporal division to consider rather than monthly divisions. Monthly
segmentation could be considered when the observed variability of time series is shorter or longer than
the 3 months length of a meteorological season.





### 4.4 Effects of length of the time series

As already stipulated under sect. 2.1, a special statistic that deviates from the normal statistic has to be applied to compute the statistical significance for N≤10. Shorter periods involve smaller N, and N is further affected by the choice of granularity. The special statistic has to be applied for trends computed on one-year averages and period < 11 years (i.e., N≤10). Note: the effect of the natural variability of a data set on trends computed on short periods will not be directly discussed here, but only the statistical effect on the trends determined for the various time series studied here.

Fig. 10 shows the effect of the reduction of the period length on the slope, the CL and the ss for the aerosol absorption coefficient dataset. The first obvious effect is that the absolute values of the slope are larger for shorter periods and there are large differences both for the individual months and meteorological seasons. Further, these large slopes for short time periods are associated with high CL and low ss. They are due to the cumulative effects of the predominant importance of the first and last years for short periods and to the low N in the time series. For the shortest period considered here (4y), the division of a daily time series into four meteorological seasons involves trends computed with N=360 (=4 years*3 months*30 days) whereas monthly trends for the same time series are computed with N=120 (=4 years*1 month*30 days). The reduction of N by a factor of three explains the larger and more variable slope values, the higher CL and the lower ss of the monthly trends compared to the meteorological season's trends. The effects due to the reduction of N are minimized by the use of daily time granularity, but they are maximized by the use of larger aggregations leading for example to N=12 and 4, respectively, for monthly aggregation (hence the tendency for increases in CL with larger aggregation in Fig. 9). It should be noted that the influence of the length of the time series is usually more important than the choice of time granularity. Also, for short time series, the yearly slopes can differ depending on the chosen temporal segmentation (see, e.g., the yearly slopes of 5y, 6y and 7y on Fig. 10). These results, then, support the standard recommendation of only computing long-term trends on time series of at least 10y.

### 4.5 Effects of the number of data points

The number of data points N in the time series is a key variable underlying the effects of the time granularity, the temporal segmentation to address seasonality and the period discussed in the previous sections. Because a long-term trend analysis is statistically sound only for time series of at least a decade in length, only decadal and multi-decadal trends are considered in this section. Figure 11 is computed using the new algorithm (e.g., Fig. 1) for all decadal trends for all time series, temporal segmentation choices and time granularities and represents the percentage of ss trend as a function of slope and N categories. Fig. 11a shows that time series with robust trends, identified by high normalized slopes, need fewer data points to reach the 95% confidence level significance than time series with less robust trends. In contrast, weaker trends, identified by low normalized slopes, need at least several hundreds or even thousands of data points to become ss. In consequence, the smallest slopes need longer periods and finer time granularities to be identified as statistically significant.

Figure 11 also clearly shows that small N leads statistically to larger normalized slopes and thus demonstrates that trends computed on short periods and with a long averaging time are usually greatly overestimated. The use of prewhitening methods with a large type 1 error will, in addition, falsely indicate



ss trends (see Sect. 4.1 and Table 3). The use of MK or TFPW-Y tests on short, highly autocorrelated and highly aggregated time series will definitely produce false positive trends with high absolute slopes.

The effects of the temporal segmentation to address seasonality and the time granularity on the confidence limits are primarily caused by the modification of N. The direct impact of N on CL as a function of slope robustness is plotted on Fig. 11b. As expected, weaker slopes and lower N lead to the largest CL with values of thousands percent of the slope for the worst cases. These high CL are not obviously related to a low ss if a prewhitening method with high type 1 error was used.

## 5. Discussion

The main effects of the various prewhitening methods on $ak_1$, the slope, the ss and the CL can be summarized as follow:

- $ak_1$ depends mostly on the intrinsic characteristics of the time series and on the choice of time granularity
- The CL intervals depend primarily on the number of data points and, thus, the length of the time series, choice of time granularity and of temporal segmentation to address seasonality.
- The ss depends mostly on the robustness of the slope, on the number of data points and on the prewhitening method.
- The slope depends mostly on the prewhitening method, with PW leading to too low slopes and MK, TFPW-Y, TFPW-WS and PW-cor resulting in absolute values of the slope that are too high, considering VCTFPW as an unbiased slope estimate.

The prewhitening methods presented here consider only the lag-1 autocorrelation. Atmospheric processes can, sometimes, be better represented by a higher order of autoregressive models with ss partial correlations at lags>1 (Table 2). These higher order lag correlations could be considered by prewhitening with the appropriate number of lags, but this was not tested during this study. Klaus et al. (2014) applied higher order autoregressive prewhitening to stable oxygen and hydrogen isotopes measured in precipitation and concluded that the ss is mostly decreased by higher order lags correlations whereas the slope is less affected. The effect of AR(2) (auto-regressive process of order 2) autocorrelation with $ak_2$= 0.2 on the type 1 and 2 errors of MK and TFPW-Y was found to be similar to strong AR(1) autocorrelation (Hardison et al., 2019) in Monte Carlo simulations, for slopes and residual variances derived from 124 ecosystem time series.

Time series with a pronounced seasonality can also exhibit an $ak_1$ seasonality. Tests were performed in order to compute $ak_1$ for the various choices of the temporal segmentation instead of on the entire time series. This variant was not further pursued due to the difficulty in applying seasonal $ak_1$, which were not always ss, leading to the application of the prewhitening method to only some of the temporal segments. These differences in the treatment of each segment yielded erratic results that could not be considered as homogeneous for a yearly trend.

The slopes computed from the various prewhitening methods for the real atmospheric data sets considered here exhibit a large spread and only studies with simulated time series are able to provide insight into the slope bias of the methods. Yue et al. (2002) shows that TFPW-Y leads to a better estimate




of the slope than PW, which systematically underestimated the real slope. Zhang and Zwiers (2004) compared the MK, PW and TFPW-WS methods for various slope and $ak_1$ strengths as well as for various

periods (30-200 years). They show that PW underestimates the slope for all slope strengths and periods for positive $ak_1$, with the biases being larger for higher autocorrelation. They also note that the biases did not decrease with the length of the time series. In contrast, they find that MK and TFPW-WS overestimate the slope for period < 200 y and high $ak_1$. In this case they showed that, while the biases are also larger for higher autocorrelation, they are significantly lower for long periods (200y), allowing calculations of

almost unbiased slope estimates. These Monte Carlo simulations used yearly time granularity so that their N corresponds to the length of the period. Their evaluation of the importance of N is not as nuanced as presented in our study in which N could be larger than the number of years in the time series for time granularities < 1 y.

The results of our study should be compared to the shortest periods (30 y) of the Zhang and Zwiers (2004)

results, where they found an underestimation of the slope by PW and an overestimation by MK and TFPW-WS. Wang et al. (2015) showed that the VCTFPW method leads to root mean square errors (RMSE) of the slope lower than the RMSE for TFPW-Y slopes for all slopes and $ak_1$ values for a time series period of 30 y. A longer period of 60 y results in lower VCTFPW RMSE only for small slopes. Finally, a recent study (Hardison et al., 2019) shows that both generalized least squares model and the Sen's slope of MK-tests

(MK and TFPW-Y) consistently overestimate the trend slope with strong $ak_1$ and short periods (up to 80% for 10 y and 21% for 20 y). The spread of the estimated slopes increases with $ak_1$ and is mediated by the length of the period. This suggests that the choice of the VCTFPW method as an unbiased estimator for time series shorter than 100 years is probably a better choice than TFPW-Y, but has to be considered in the context of the CL size in order to obtain a better estimate of the real long-term trend.

All the simulation studies described above report slope per year based on yearly aggregated time series. Their number of data points corresponds then to the time series length. In contrast, N as defined in this study, could be much larger for an equivalent time series length as we considered data aggregations between 1d to 1y. The shortest simulated periods were 10 y (Hardison et al., 2019, Yue and Wang, 2004, Hamed, 2009), 20 y (Yue et al., 2002), 25 y (Bayazit and Önöz, 2007) and 30 y (Zhang and Zwiers, 2004,

Wang et al., 2015). All the recommendations of these authors about erratic results for "short periods" always concern decadal or even multi decadal trends and are, consequently, even more relevant for trend results for periods shorter than 10 y.

Based on the results presented in this study as well as the findings from the literature referenced above, the following recommendations can be made:

● A prewhitening method must be used on time series when $ak_1^{data}$ is ss.
● The seasonal MK-test must be used on time series with a clear seasonal cycle. The chosen temporal segmentation to address seasonality for the MK-test has to be compatible with the observed seasonality of the time series.
● Finer time granularities should be used in order to maximize the number of data points and will

yield smaller confidence limits and larger ss. The choice of the time granularity must also be compatible with the observed seasonality of the time series.
● Periods shorter than 10 y must be handled with great caution and periods shorter than 8 y should not be used for long-term trend analysis.



- When describing trend results the sign of the slope should not be mentioned if it is not ss, because not ss trends cannot, by definition, be distinguished from zero trends. Moreover, not ss trends have a larger dependency on how the trends are computed (time granularity, period, prewhitening method, temporal segmentation to address seasonality,…).
- In the presence of ss lag-1 autocorrelation, either PW and TFPW-Y together or TFPW-WS should be used to assess statistical significance. MK, TFPW-Y alone and VCTFPW lead to a high number of false positives.
- The slope should be corrected in order to take into account the effect of the prewhitening on the mean and the variance of the time series. We recommend the VCTFPW method to eliminate slope biases, at least for time series shorter than 30 y.
- In presence of ss trends, the confidence limits must also be considered in order to assess the uncertainty in the slope.

## 6. Conclusion

Several prewhitening methods including solely prewhitening, the trend-free prewhitening from Yue et al. (2002) and from Wang and Swail (2001) as well as the variance-corrected trend-free prewhitening method of Wang et al. (2015) were tested on seven time series of various in-situ and remote sensing atmospheric measurements. Consistent with the literature, the use of MK, TFPW-Y and VCTFPW results in a large amount of false positive results while TFPW-WS results in less than 2% of false positives. The power of the test is good for all the applied MK-tests for the time series considered here.

The effect of the choosing time granularities from 1 day to one year was also evaluated since a common way to overcome the autocorrelation problem is to average time series to a coarser time granularity. It was found that the $ak_1^{data}$ could remain ss up to at least monthly granularity and was sometimes still ss for yearly averages. Finer time granularities exhibit higher $ak_1^{data}$ leading to a larger difference of the estimated slope by the various prewhitening methods. MK, TFPW-Y, TFPW-WS and PW-cor result in the largest absolute values of the slope and PW the smallest. VCTFPW slopes are found between these two extremes. The confidence limits are much broader for coarser time granularities and the ss is lower, so that ss at the 95% confidence level is rarely achieved. The main impact of keeping a fine time granularity is that it allows computation of the trends on a high number of data points, which improves the power of the test and decreases the uncertainties in the slope.

Since all the time series studied exhibited clear seasonal cycles, two temporal segmentations (12 months and 4 meteorological seasons) were tested for the seasonal MK-test. The segmentation into four meteorological seasons resulted in more homogeneous trends among the segments, a necessary condition to compute yearly trends. The division into meteorological seasons also resulted in a higher number of data points available in each temporal segment relative to division into monthly segments. No systematic effect of the choice of temporal segment on the slope was observed and the difference between temporal segment choices was always much lower than the differences among the prewhitening methods.



Finally, a new algorithm was proposed combining several prewhitening methods to obtain a better estimate of trend and statistical significance than would be achieved with any individual prewhitening method. PW and TFPW-Y were used to compute the statistical significance of the trend and VCTFPW was applied to estimate the slope. This approach takes advantage of the low type 1 errors of PW, the high test power of TFPW-Y and the less biased slope estimated by VCTFPW.

## Acknowledgements

The authors would like to thank Patrick Sheridan (NOAA) for mentoring and providing the Bondville data, Derek Hageman (University of Colorado) for programming efforts in data acquisition and archiving, and the on-site technical staff from the Illinois State Water Survey for their long-term support and care for the instrumentation.

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





**Tables**


*Table 1: Advantages and disadvantages of the MK-test and of the various prewhitening methods.*

| Method | How it works | Advantages/Disadvantages |
|---|---|---|
| MK | • Applied on the data without modification | • High type I error<br>• High test power<br>• slope increased by $ak_1^{data}$ |
| PW<br>(Kulkarni & von Storch, 1995) | • Remove the autocorrelation | • Low type I error<br>• Low test power<br>• Smaller absolute slope |
| PW-cor | • Remove the autocorrelation<br>• Preserve the slope | • Low type I error<br>• Low test power<br>• Similar slope as MK |
| TFPW-Y<br>(Yue et al., 2002) | • Remove the slope<br>• Remove the autocorrelation<br>• Add the trend | • High type I error<br>• High test power<br>• Larger absolute slope |
| TFPW-WS<br>(Wang & Swail, 2001) | • Apply TFPW iteratively until $ak_1^{detr\text{-}prew}$ and the slope stay constant:<br>  ➢ Remove the autocorrelation<br>  ➢ Compute the slope<br>  ➢ Remove the trend from the original data<br>• Remove the final $ak_1^{detr\text{-}prew}$ | • Low type 1 error<br>• High test power<br>• Similar slope as MK |
| VCTFPW<br>(Wang, 2015) | • Remove the trend<br>• Remove the autocorrelation<br>• Correct the variance similar to initial variance<br>• Add the trend with corrected slope | • Middle type I error<br>• Medium test power<br>• Unbiased slope estimate |





*Table 2: Description of the time series: time series with units, monitoring station, period, instrument type, original granularity, ranges (1 and 99 percentiles (*1%ile *and* 99%ile*)), mean, median and standard deviation (STD), lag-1 autocorrelation of the observations (ak$_1^{data}$) and number of ss partial autocorrelations for the 10 y period (order), number of data in the 10y period (N) and reference.*

| Time series | Station | Period | Instrument | Granularity | 1%ile 99%ile | Mean Median STD | $ak_1^{data}$ order | N | reference |
|---|---|---|---|---|---|---|---|---|---|
| Aerosol scattering coef. [Mm$^{-1}$] | BND | 1995-2018 | TSI Nephelometer | 1 h | 6.57 167.80 | 43.51 33.04 33.85 | 0.60 2 | 3485 | Sherman et al., 2015 |
| Aerosol absorption coef. [Mm$^{-1}$] | BND | 1995- 2018 | PSAP and CLAP | 1 h | 0.51 11.06 | 3.40 2.85 2.30 | 0.53 2 | 3431 | Andrews et al., 2019 |
| Aerosol number concentration [cm$^{-3}$] | BND | 1995- 2018 | CPC | 1 h | 283 11636 | 4139 3674 2517 | 0.58 2 | 2979 | Laj et al., 2020 |
| Aerosol optical depth | PAY | 2006-2015 | PFR | 1 h | 0.025 0.285 | 0.126 0.113 0.064 | 0.72 2 | 641 | Nyeki et al., 2019 |
| Tropopause level [m] | PAY | 1958-2018 | Radio-sonde | 12 h | 7540 14660 | 11178 11280 1425 | 0.70 2 | 3636 | Brocard et al., 2013 |
| Zero degree level [m] | PAY | 1958-2018 | Radio-sonde | 12 h | -859 4437 | 2333 2457 1208 | 0.89 3 | 3640 | Brocard et al., 2013 |
| Water Vapor Mixing ratio [g/kg] | PAY | 2009-2018 | Ralmo Lidar | 0.5 h | 1.41 11.88 | 5.90 5.57 2.63 | 0.88 3 | 2868 | Hicks-Jalali et al., 2019 |

PSAP=Particle Soot Absorption Photometer, CLAP=Continuous Light Absorption Photometer, CPC=Condensation Particle Counter, PFR=Precision Filter Radiometer.

*Table 3: Percent of false positives and false negatives for all data sets relative to a reference test for the MK-tests and prewhitening methods for periods of at least 10y (decadal trends) or smaller than 8y.*

| Period | Type of error | MK | TFPW-Y | TFPW-WS | PW | PW-cor | VCTFPW |
|---|---|---|---|---|---|---|---|
| ≥ 10y | False positive | 33.5 | 37.4 | 1.7 | reference | 0.0 | 48.5 |
| N=2185 | False negative | 5.3 | reference | 0.2 | 0.2 | 0.2 | 26.1 |
| < 8y | False positive | 19.8 | 14.3 | 1.1 | reference | 0.0 | 44.9 |
| N=1045 | False negative | 7.0 | reference | 0.3 | 0.0 | 0.3 | 36.6 |

*Table 4: Percentage of yearly trends with homogeneous temporal segments as a function of the type of segment (month or season), of the prewhitening method and of the length of the periods based on all seven time series considered in this study.*

| Period | Method | Months | Meteorological seasons |
|---|---|---|---|
| ≥ 10y | VCTFPW | 26.1 % | 80.0 % |
| N=115 | TFPW_Y | 25.2 % | 86.1 % |
| < 8y | VCTFPW | 5.5 % | 74.5 % |
| N=55 | TFPW_Y | 5.5 % | 80 % |




**Figures**

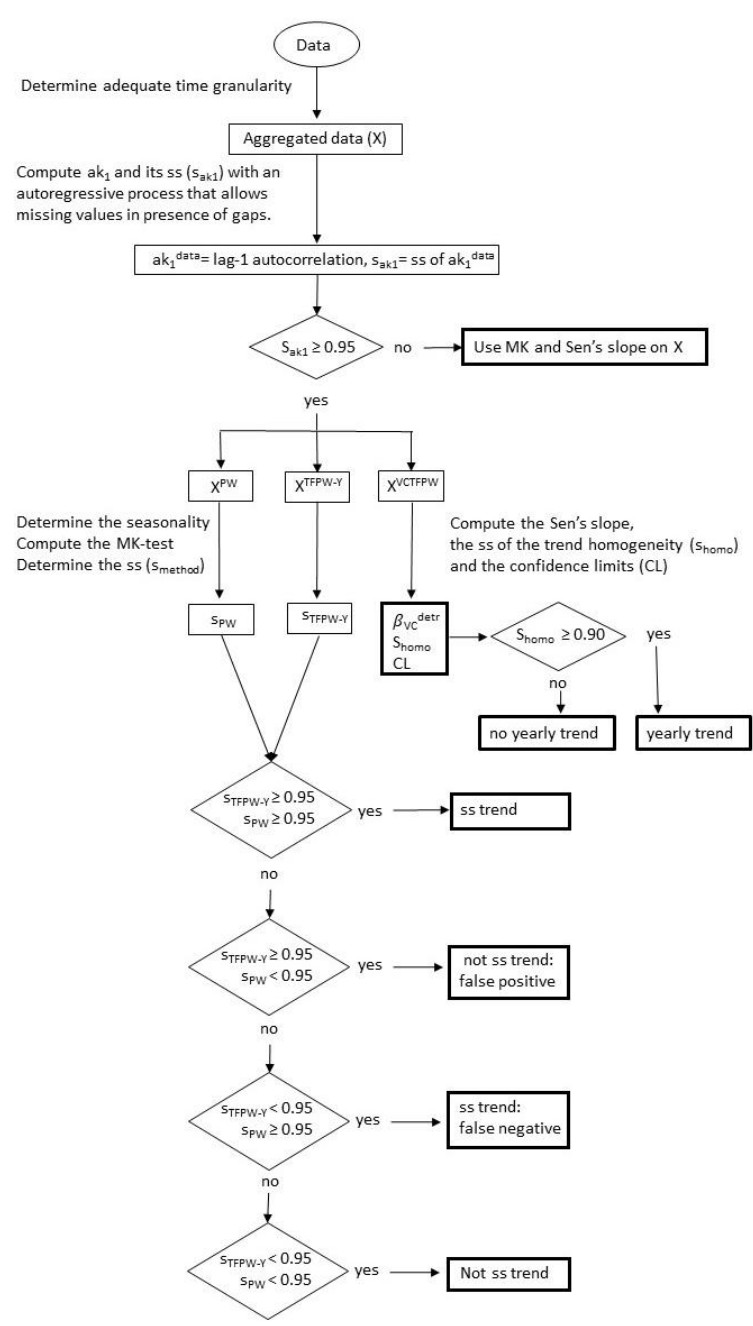

*Figure 1: Scheme of the new algorithm*



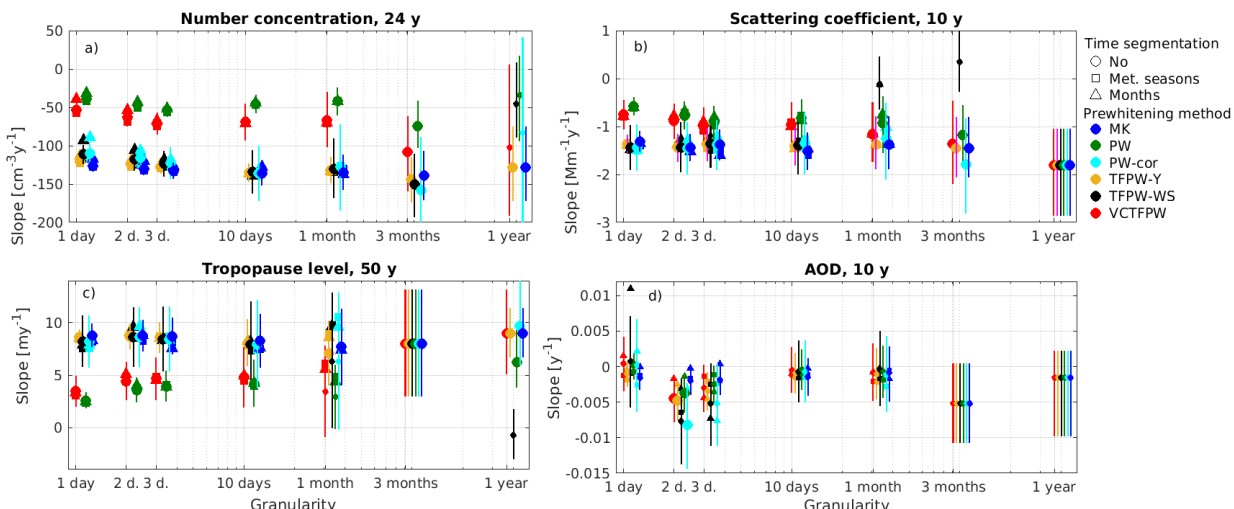

*Figure 2: Slope and confidence limits as a function of the time granularity for MK and the five prewhitening methods (indicated by colors) and for various temporal segmentation choices (indicated by symbols) for a) the aerosol number concentration for the 24 y period, b) the aerosol absorption coefficient for the 10 y period, c) the tropopause level altitude for the 50 y period, and d) the AOD for the 10 y period. Larger symbols indicate ss trends and confidence limits are plotted only without time segmentation for clarity purposes.*

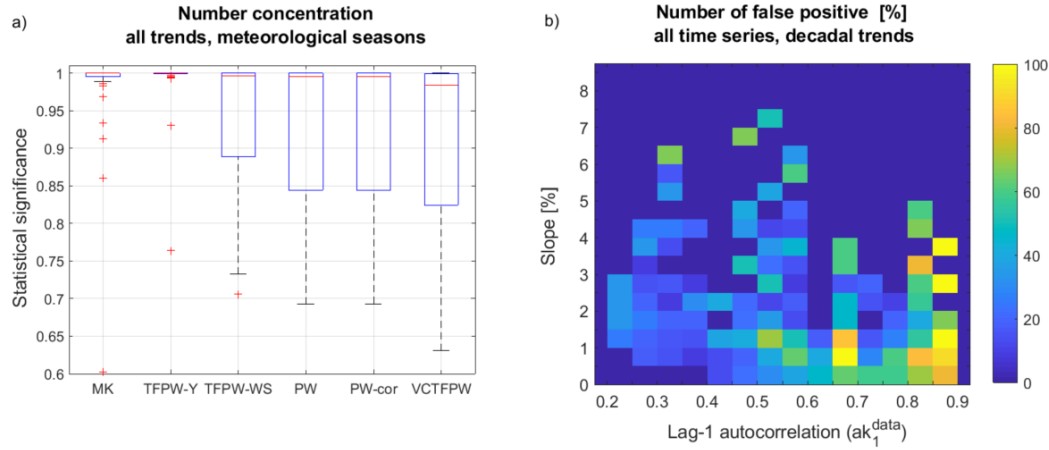

*Figure 3 a) Statistical significance of slopes as a function of the prewhitening methods for the aerosol number concentration for the yearly trends computed from four meteorological seasons, for all periods (5y to 24y) and all time granularities (40 trends). The median is represented by the red line, the boxes are*





*the 25% and 75% percentiles, the whiskers the 0.7 and 99.3 percentiles and the red plus signs the outliers.*
*Some outliers are not on the figure for purposes of clarity.*

*b) number of TFPW-Y false positives as a function of $ak_1^{data}$ and slope categories for all the computed trends of all time series for decadal periods. Categories with less than 3 points are not plotted.*

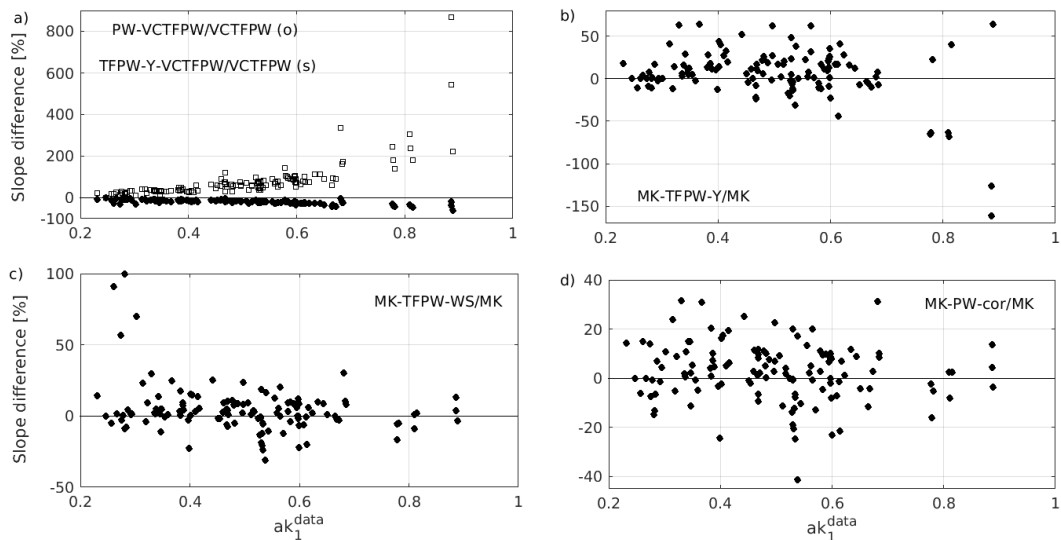


*Figure 4: Slope differences as a function of $ak_1^{data}$ from the original data for all datasets and periods and for meteorological season time segmentation: a) PW minus VCTFPW slope (filled dots) and TFPW-Y minus VCTFPW slope (open squares) normalized by the VCTFPW slope, b) MK slope minus TFPW-Y slopes, c) MK minus TFPW-WS slopes and d) MK minus PW-cor slopes. The slope difference in b) c) and d) are*
*normalized by MK slope. Not ss trends (PW taken as reference) are not plotted since the slopes cannot be distinguished from zero trend. Note the different y-axis ranges on these plots.*



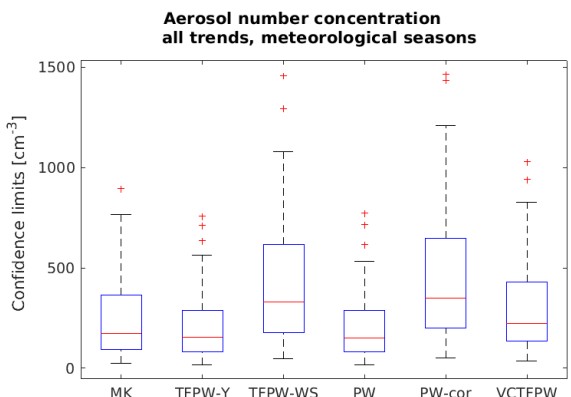

*Figure 5: Distribution of the confidence limit intervals of the slope for the trend in aerosol number*
*concentration for all periods (5y-24y) and time granularities as a function of the method for the*
*meteorological seasons temporal segmentation. Box-whisker plotting as described for figure 3a.*

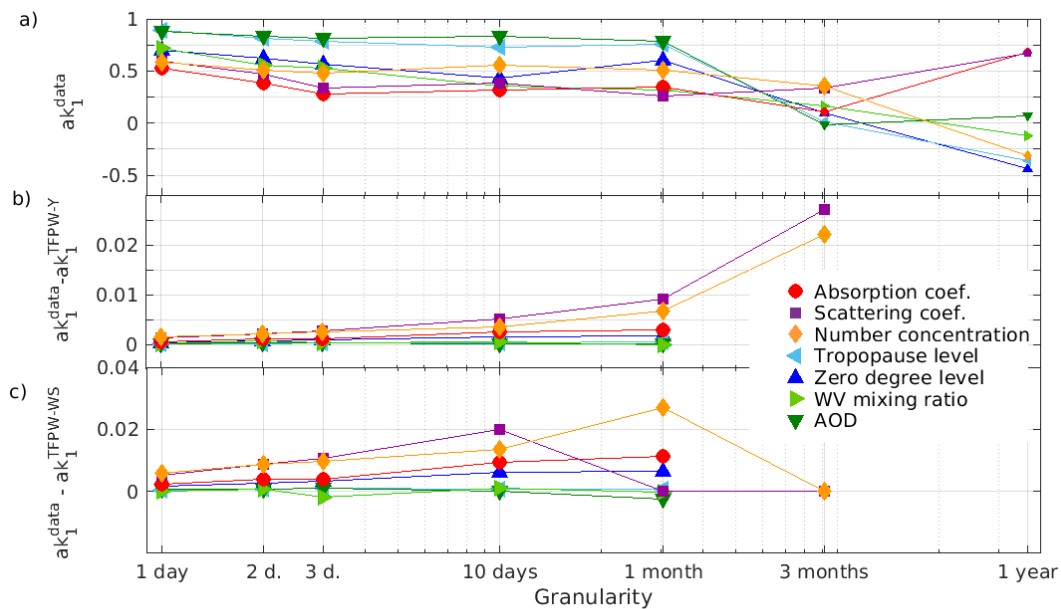

*Figure 6: a) Lag-1 autocorrelation ($ak_1^{data}$) of the original data as a function of the time granularity for the*
*10 y time series of all time series, bigger symbols correspond to ss $ak_1^{data}$ b) $ak_1$ difference between the*
*original data and the TFPW-Y data, and c) $ak_1$ difference between the original data and the TFPW-WS*
*data. For b) and c) only ss cases are plotted because prewhitening methods are not applied when $ak_1$ is*
*not ss.*





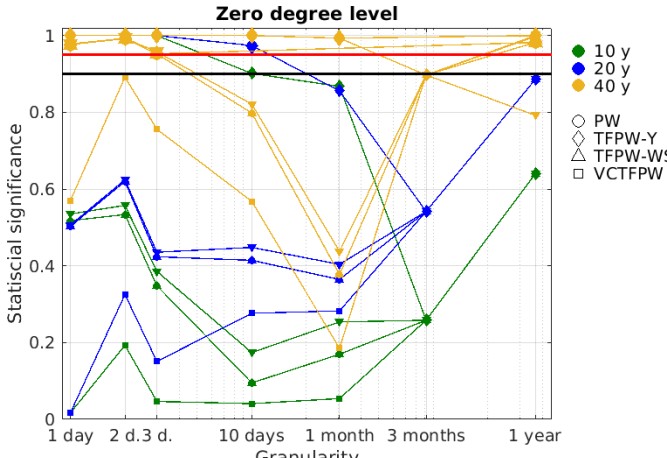


*Figure 7: Statistical significance of the trends as a function of the time granularity and prewhitening methods for the zero degree level time series for 10y, 20y and 40y periods without temporal segmentation to address seasonality. The horizontal red and black lines correspond to the threshold of 95% and 90% confidence level, respectively, and ss trends are also emphasized by bigger symbols.*


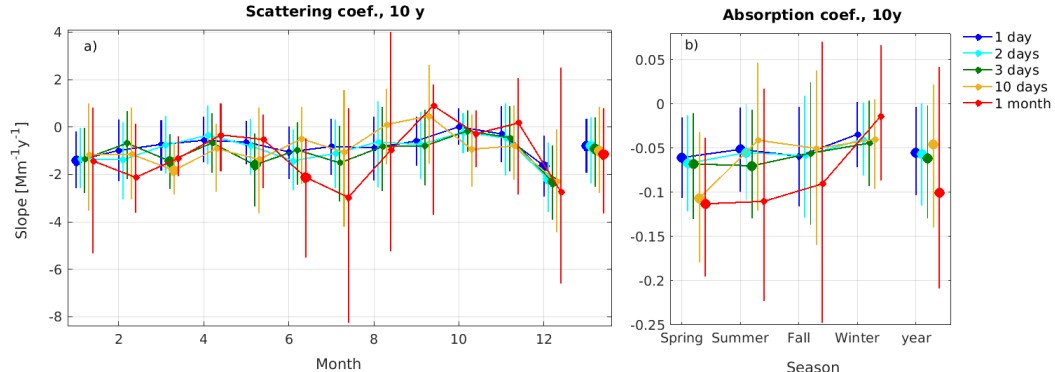

*Figure 8: VCTFPW slope as a function of the time granularity for the division of the time series into a) 12 months for the 10 y aerosol scattering coefficient and b) into four meteorological seasons for the 10 y*

*aerosol absorption coefficient. Larger symbols indicate statistically significant slopes computed from the new algorithm.*



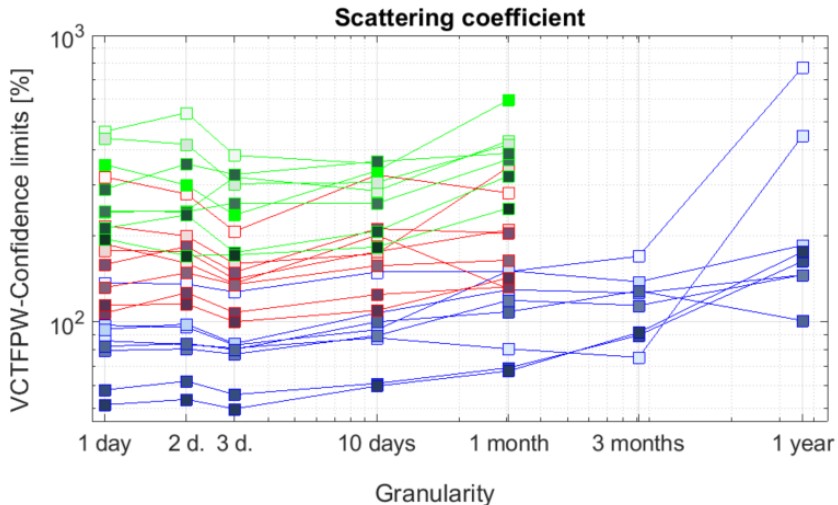

Figure 9: Confidence limits of VCTFPW as a function of the time granularity for various lengths of the
aerosol scattering coefficient time series. Blue represents for no consideration of seasonalities; red
represents division into 4 meteorological seasons and green represents division into 12 months. The color
shading corresponds to the length of the period from 5 y (lightest) to 24 y (darkest).

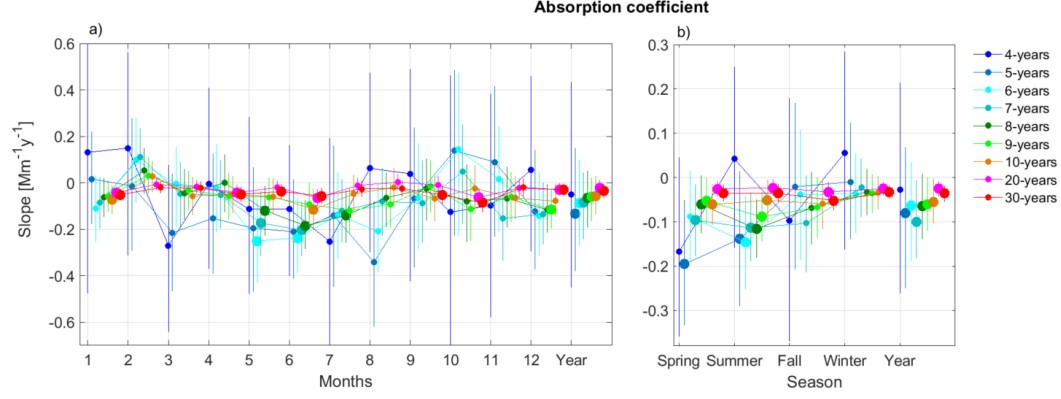


Figure 10: VCTFPW slopes and CL as a function of various periods for the daily aerosol absorption
coefficient for the division of the time series into a) 12 months and b) four meteorological seasons. Colors
represent time period lengths and bigger symbols represent ss trends.



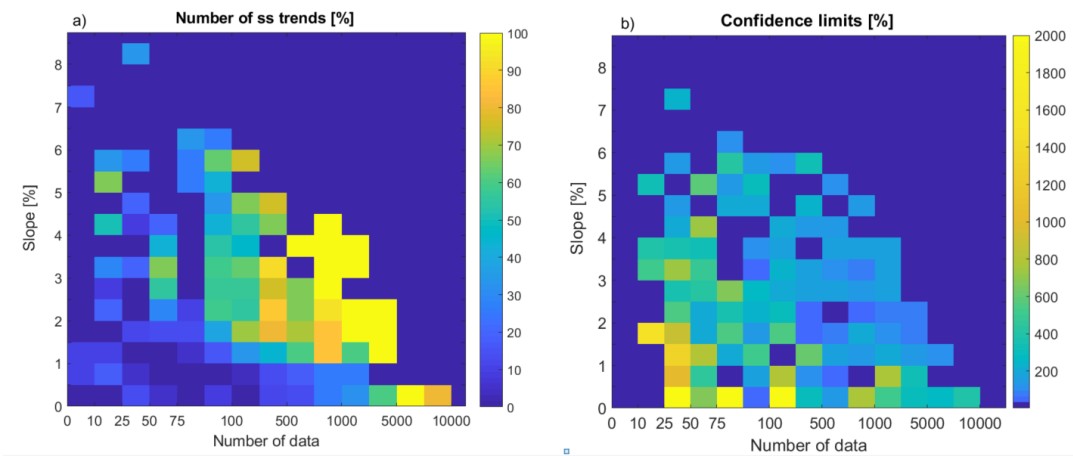


*Figure 11: a) The percentage of ss trends from the new algorithm (sect. 2.3) and b) mean confidence limits normalized by the slope as a function of normalized slope and N categories for all computed trends with period of at least a decade. The slopes are binned regularly (bin size = 0.5%) but N categories are irregular. Cells with less than 3 results were discarded in panel a).*