# Peer review of "Effects of the prewhitening method, the time granularity, and the time segmentation on the Mann-Kendall trend detection and the associated Sen's slope"

_Atmospheric Measurement Techniques, 2020_

## Referee Comment (RC1) · Wenpeng Wang (Referee) · 15 Jul 2020

**Effects of the prewhitening method, the time granularity and the time segmentation on the Mann-Kendall trend detection and the associated Sen's slope**

**By Martine Collaud Coen, Elisabeth Andrews, Alessandro Bigi, Gonzague Romanens, Giovanni Martucci and Laurent Vuilleumier**

**Submitted to Atmospheric Measurement Techniques amt-2020-178**

**REFEREE REPORT**

**General comments**

The authors propose a new algorithm of trend analysis on autocorrelated meteorological data via incorporating the merit of three prewhitening techniques. The effect of time granularity, time segmentation and time series length on trend analysis are also evaluated on the basis of real meteorological observations.

The proposed algorithm is a good trial to purse the ideal goal of trend detection methods, that is high power with controllable Type I error, and accurate slope estimates. I think this algorithm is practically sound.

But still I have reservations about some statements in this paper. The manuscript and the quality of figures should be improved before it is formally published.

**Specific comments**

1. Line 120-122. "These approaches (variance correction approaches) appear not able to preserve the significance level and the power of the MK-test in the case of correlated time series with a trend"

Comment: Both the variance correction approach and the prewhitening approach can preserve the pre-assigned significance level when there is no trend. Because detecting trends with known statistical confidence is the primary goal of trend analysis, either on independent data or autocorrelated data. The power of trend identification may be different for distinct methods.

2. Line 139-140: *it* (*PW method*) *reduces the power of the test due to an over-/underestimation of*  $ak_1^{data}$  *in the case of a positive/negative trend.*

Comment: The existence of real trend, either positive or negative, can lead to an overestimation of lag-1 autocorrelation coefficient.

3. Comment on Line 151-164: The brief introduction on the TFPW-WS method (Wang and Swail's 2001) includes some mistakes. I suggest rephrase this paragraph.

The original idea of Wang and Swail's (2001) was intended to implement the MK test on the prewhitened series, rather than on the prewhitened detrended series, as it was given by Eq.(8). If the prewhitened series are detrended, then we will never identify any trends.

The critical value to stop iteration should be a tiny number, e.g. 0.0001, instead of 0.05.

The primary consideration of iteration procedure was to mitigate the adverse effect of trend on the accuracy of lag-1 autocorrelation coefficient estimate.

4. Line 164-167: The PW-cor method refers to the preliminary step of the first iteration in the TFPW-WS method and consequently corrects the prewhitened data by the same factor. To the knowledge of the authors, this PW-cor method is not referenced in the literature but is a potential method tested in this study.

Comment: After rephrasing the TFPW-WS method, please describe the PW-cor method more clearly.

5. Line 184-185: VCTFPW preserves to some extent the power of the test, but only mitigates the type 1 errors.

Comment: Similar to other prewhitening methods, the VCTFPW method mitigates the inflationary type 1 errors raised by autocorrelation as its priority. Then the method preserves the power of the trend test to some extent.

6. Line 203-204: If PW is ss but TFPW-Y is not, then the trend is considered as a false negative due to the lower test power of PW and the trend has to be considered as ss.

Comment: If we consider the trend to be statistically significant, then we cannot say the detected trend is a false negative result. It is illogical to report a trend and meanwhile state this is an error.

Figure 1 should be revised accordingly.

7. Line 225-228: Trend analyses were applied on several periods. For all the data sets, a 10-year period is considered first and then further possible multi-decadal periods up to 60 y for the radio-sounding time series. For the in-situ aerosol properties, tests with 4 to 9 y periods are also computed in order to illustrate the problems of trend analysis on very short time series.

Line 781-782: lag-1 autocorrelation of the observations (ak1data) and number of ss partial autocorrelations for the 10y period (order), number of data in the 10y period (N) and reference.

Comment: I think in this section "3 Experimental", the authors should clarify how to analyze the measure data, in order to support the coming results. The meaning of "*a 10-year period*" or "*multi-decadal periods up to 60 y*" are unclear and obscure.

In table 2, the meaning of "number of ss partial autocorrelations for the 10y period (order), number of data in the 10y period (N) and reference" is unclear either.

8. Line 275-276: *CL of MK*, *PW and TFPW-Y*, *which remove the lag-1 autocorrelation without compensation for the mean values and the variances*...

Comment: Does that mean "mean and variances of the slope estimate"?

9. Line 278-280: The ss often decreases for coarser time granularities occasionally leading to not ss trends for some of the prewhitening methods. PW, TFPW-WS and VCTFPW methods become not ss at finer time granularities than TFPW-Y and MK due to their lower number of false positives.

Comment: It's hard to identify the relationship between the significance of trend and the time granularity from Fig. 2.

10. Line 281-282: The discrepancies between prewhitening methods are larger than the discrepancies that occur when different temporal segmentations (months or meteorological seasons) are applied.

Comment: Fig.2 does not support this finding.

11. Line 284-285: the similarity of MK slopes with TFPW slopes.

Line 350-352: Due to the detrending procedure, the absolute values of the TFPW-Y

slope are larger than the PW slopes and similar to the MK slope values (Fig. 2), even if a tendency to have larger TFPWY than MK slopes are observed.

Line 367: TFPW-Y slopes tend to be larger than MK slopes (Fig. 4b), with larger differences at high ak1 data leads.

Comment: If my understanding is right, the MK and TFPW-Y should yield exactly the same slope of trend. The MK test does not estimate the slope of trend directly. It usually reports the magnitude of trend by the use of Sen's slope. The TFPW-Y also estimates Sen's slope as its first step. It will reinstall this trend to the prewhitened series without any modification. So these two slopes should be equal to each other.

12. Line 285-287: For example, the number of data points in the AOD time series (about 65 per year) induces higher CL for time granularities finer than the measurement frequency (about 10 days).

Line 372-373: Removing the lag-1 autocorrelation increases the variance, but decreases the mean.

Line 391-395: The spread of the slopes of the aerosol number concentration for the one-year aggregation on Fig. 2c shows that the yearly data still have a ss ak1 data for the longest periods of 20 and 24 years (see similar cases in Fig. 2). For shorter periods (5 to 9 years), the ak1 data decreases rapidly for averaging longer than 10 days and even becomes negative for yearly averages.

Comment: These sentences are difficult to be understood. Please rephrase.

13. Line 294-296: The yearly trend was computed for all periods (from 5y to 24y) at all considered time granularities (1 day to 1 month for the meteorological season temporal segmentation), leading to 40 trends.

Comment: Please clarify what is the 40 trends?

14. Line 323-325: PW is used as the reference for false positives because it is the prewhitening method with the lowest type 1 error, while TFPW-Y is the reference for false negatives because it is the most powerful test.

Comment: It's inappropriate to state that the TFPW-Y is the most powerful test. The TFPW-Y tends to report significant trends at the expense of committing high type 1 error. This finding has been verified by many literatures. So we can say the TFPW-Y tends to identify significant trends more frequently than other methods, but we cannot

say it is the most powerful test.

15. Line 338-342: For the time series considered in this study, the following conclusions can be made: 1) PW performs very well with an almost vanishingly small ( $\leq 0.3\%$ ) number of false negatives and the ss of PW-cor is similar to that for PW; 3) VCTFPW has a very high type 1 and 2 errors and should consequently not be used to determine the ss; and 4) it is not possible to determine whether MK or TFPW-Y is the most powerful method.

Line 786-788: *Table 3*

Comment: The three conclusions made here do not align with the consensus about the prewhitening method among the community. I suggest to recheck the results.

1) The PW tends to overestimate the lag-one autocorrelation coefficient without trend removal, see Hamed (2009). In addition, the PW reduces a portion of real trend, see Yue and Wang (2002). That's the reason why Yue et al. (2002) suggest to remove trend before whitening. So if the TFPW-Y is the reference for false negatives, the PW is less likely to miss only 0.2% significant trends.

3) As it was stated by the authors, e.g. Line 265-266, Table 1, Figure 4(a). The VCTFPW slopes lies between the TFPW and the PW slope values. So no matter one takes the PW or the TFPW-Y as the reference, the VCTFPW is less likely to commit the highest error among all the prewhitening methods.

4) For the autocorrelated data, the MK and TFPW-Y are not really powerful method. They only tend to report significant trends more frequently than other PW methods. However, both of them commit high type I error as a price.

I have to say, the above opinions are given by Monte-Carlo simulation results. They may not suitable to every real-world series. This study deals with measured data. So I suggest to recheck your results again.

**16. Line 345: The slope of the trend is always enhanced by the positive ak1data.**

Comment: I think it should be "the slope estimates of the trend is influenced by the positive lag-one autocorrelation". The autocorrelation increases the difficulty of an accurate slope estimation. But it does not increase or decrease the real slope of the trend.

17. Line 628-629: Consistent with the literature, the use of MK, TFPW-Y and

VCTFPW results in a large amount of false positive results while TFPW-WS results in less than 2% of false positives.

Comment: After recheck your results, e.g. table 3, this conclusion should be revised accordingly.

18. Line 637: The confidence limits are much broader for coarser time granularities and the ss is lower.

Comment: Fig. 8 supports this conclusion but Fig. 10 does not. As the time granularity becomes coarser, the confidence limits are much narrower in Fig. 10.

19. Comment on Figure 2: it is hard to distinguish the time segmentation.

20. Comment on Figure 7: it is not easy to identify different PW methods.

21. Comment on Figure 8 and 10: it is unclear how to analyze the slope of trend as well as the confidence limit within each time segmentation. It should be well explained.

22. I suggest to improve the quality of the figures, to make them self-explaining.

**Technical corrections**

Line 109. Zwang and Zwiers (2004) does not given in the reference list.

Line 168. I think the correct citation about the VCTFPW method should be "Wang, W., et al., 2015. Variance correction pre-whitening method for trend detection in auto-correlated data. Journal of Hydrologic Engineering, 04015033. doi:10.1061/(ASCE)HE.1943-5584.0001234."

Line 272: "aerosol absorption coefficient" should be "aerosol scattering coefficient".

Fig 2 Caption: "*Scattering coefficient, 10y*" and "Tropopause level, 50y". Should it be "24y" and "60y" ?

**References**

- Hamed, K.H., 2009. Enhancing the effectiveness of prewhitening in trend analysis of hydrologic data. Journal of Hydrology, 368(1-4): 143-155.
- Yue, S., Pilon, P., Phinney, B., Cavadias, G., 2002. The influence of autocorrelation on the ability to detect trend in hydrological series. Hydrological Processes, 16(9): 1807-1829.
- Yue, S., Wang, C.Y., 2002. Applicability of prewhitening to eliminate the influence of serial correlation on the Mann-Kendall test. Water Resources Research, 38(6): 1068.

---

## Referee Comment (RC3) · Anonymous Referee #1 · 17 Jul 2020

This manuscript is well written and is an important contribution for more comprehensive trend analysis of atmospheric composition data. The work is robust with very good analysis and discussions of the different effects on the trend results using various prewhitening methods in addition to MK without prewhitening. It is very well appreciated the clear guidelines for choosing methods and approaches for assessing long term trends.

I will recommend the paper to be published as it is. I have only some small comments/questions which you may consider:

[Figure]

Line 125. why is negative autocorrelation rare in atmospheric processes? Maybe explain a bit more the reasons and differences between negative and positive autocorrelation and/or give a reference.

Line 272. Why is aerosol number concentration behaving different than the other components regarding the effect of granularity, i.e. the ss remain until the one-year aggregation?

Fig8 and paragraph 429-436. Here you compare the difference in granularity of monthly and seasonal data. Why use different data (scattering contra absorption)? To illustrate the difference in granularity it would have been more logic to use same dataset?

Fig10 and paragraph 493-509. Not sure if I understand how the data selection has been done. Do all the periods contain the whole time serie? I.e 10years contain 3x10years data set if the time serie is totally 30 years. I assume you somehow taken into account that the actual trend for the whole period will effect the results. Not homogeneous trend over a 30 year period. But why is it then so few data points for the 4 year trend, I,e N=360 and 120 for monthly and seasonal trends?

The new algorithm applied. Is that made available? The scheme sketched in Figure 1 is not very easy to use for others to apply the method. It is recommended that the authors upload the scripts for others to use and adopt if possible.

---

## Author Comment (AC1) · 14 Oct 2020

**Responses to Wenpeng Wang**

The authors thank Wenpeng Wang for his detailed review of the manuscript, and for all the comments and suggestions allowing a clear improvement of the paper. The line numbers correspond to the manuscript submitted to AMTD.

**Responses to specific comments:**

**General comments**

The authors propose a new algorithm of trend analysis on autocorrelated meteorological data via incorporating the merit of three prewhitening techniques. The effect of time granularity, time segmentation and time series length on trend analysis are also evaluated on the basis of real meteorological observations.

The proposed algorithm is a good trial to purse the ideal goal of trend detection methods, that is high power with controllable Type I error, and accurate slope estimates. I think this algorithm is practically sound.

But still I have reservations about some statements in this paper. The manuscript and the quality of figures should be improved before it is formally published.

**Specific comments:**

1. Line 120-122. "These approaches (variance correction approaches) appear not able to preserve the significance level and the power of the MK-test in the case of correlated time series with a trend"

Comment: Both the variance correction approach and the prewhitening approach can preserve the pre-assigned significance level when there is no trend. Because detecting trends with known statistical confidence is the primary goal of trend analysis, either on independent data or autocorrelated data. The power of trend identification may be different for distinct methods.

Your comment is completely right. The sentence was consequently improved: "These approaches appear to preserve the pre-assigned significance level and the power of the MK-test in the absence of trend but not in the case of correlated time series and in the presence of a trend (Yue et al., 2002; Blain, 2013)."

2. Line 139-140: *it (PW method) reduces the power of the test due to an over-*/underestimation of ak1data in the case of a positive/negative trend.

Comment: The existence of real trend, either positive or negative, can lead to an overestimation of lag-1 autocorrelation coefficient.

The sentence was modified for clearity: "This PW method results in a low amount of type 1 errors, but the existence of real trends, either positive or negative, can lead to an over-/underestimation of  $ak_1^{data}$ , which will reduces the power of the test."

3. Comment on Line 151-164: The brief introduction on the TFPW-WS method (Wang and Swail's 2001) includes some mistakes. I suggest rephrase this paragraph.

The original idea of Wang and Swail's (2001) was intended to implement the MK test on the prewhitened series, rather than on the prewhitened detrended series, as it was given by Eq.(8). If the prewhitened series are detrended, then we will never identify any trends. The critical value to stop iteration should be a tiny number, e.g. 0.0001, instead of 0.05. The primary consideration of iteration procedure was to mitigate the adverse effect of trend on the accuracy of lag-1 autocorrelation coefficient estimate.

The authors do agree with the proposition of the reviewer and the manuscript was modified consequently:" The original idea of Wang and Swail's (2001) was intended to implement the MK test on the prewhitened series, rather than on the prewhitened detrended series, as it was given by Eq.(8). If the prewhitened series are detrended, then trends will not be identified. Wang and Swail's (2001) propose an iterative TFPW method to mitigate the adverse effect of trend on the accuracy of the lag-1 autocorrelation estimate. This iterative procedure consists of: i) removing  $ak_1^{data}$  from the original time series and correcting the prewhitened data for the modified mean (eq. 5); ii) estimating the Sen's slope  $\beta^{prew}$  on the prewhitened data  $A_{cor,t}^{prew}$ ; iii) removing the trend ( $\beta^{prew}$ ) estimated on the PW data from the original data to obtain a prewhitened detrended time series  $A_{cor,t}^{detr}$  (eq. 6); and iv) applying iteratively i-iii until the  $ak_1$  and slope differences become smaller than a proposed tiny threshold of 0.0001 (eq. 7)."

All the TFPW-WS trends have been recomputed with the threshold of 0.0001 without any marked differences in the results. The following sentence was then added at line 160: "Note that the use of a higher threshold up to 0.05 does not significantly modify the results obtained on the considered time series."

4. Line 164-167: The PW-cor method refers to the preliminary step of the first iteration in the TFPW-WS method and consequently corrects the prewhitened data by the same factor. To the knowledge of the authors, this PW-cor method is not referenced in the literature but is a potential method tested in this study.

Comment: After rephrasing the TFPW-WS method, please describe the PW-cor method more clearly.

The PW-cor is now described with more details: "The preliminary step of the first iteration in the TFPW-WS method (removing  $ak_1^{data}$  from the original time series and correcting the prewhitened data for the modified mean eq. (5)) corresponds to the standard PW method but with the same correction factor ensuring a similar trend between the prewhitened and the original time series. This method called PW-cor is, to the knowledge of the authors, not referenced in the literature but is a potential method tested in this study."

5. Line 184-185: VCTFPW preserves to some extent the power of the test, but only mitigates the type 1 errors.

Comment: Similar to other prewhitening methods, the VCTFPW method mitigates the inflationary type 1 errors raised by autocorrelation as its priority. Then the method preserves the power of the trend test to some extent.

The authors consider that the first "priority" of the VCTFPW method is to preserve the value of the slope and that this priority has also some effects on the type 1 and type 2 errors. The manuscript was modified: "*Statistical simulations by Wang* (2015) showed that this new variance corrected prewhitening method (VCTFPW) leads to more accurate slope estimators, tends to mitigate the inflationary type 1 errors raised by autocorrelation and preserves to some extent the power of the test."

6. Line 203-204: If PW is ss but TFPW-Y is not, then the trend is considered as a false negative due to the lower test power of PW and the trend has to be considered as ss. Comment: If we consider the trend to be statistically significant, then we cannot say the detected trend is a false negative result. It is illogical to report a trend and meanwhile state this is an error.

Figure 1 should be revised accordingly.

The referee is right. If PW is ss but TFPW-Y is not, this is not a TFPW-Y false negative but a PW false positive and this has to be considered as not ss. Figure 1 was changed accordingly and the manuscript is revised:

- Lines 203-204: "If TFPW-Y is ss but not PW, the trend is considered as a TFPW-Y false positive due to the too high type 1 errors of TFPW-Y and the trend has to be considered as not ss. If PW is ss but TFPW-Y is not, then the trend is considered as a PW false positive and the trend has to be considered as not ss."
- § 4.1: lines 322-342: "To obtain a better view of the weakness of each MK-test, the percentage of false positives taking each of the prewhitening method as reference are reported in Table 3 for all the datasets. PW-cor has by definition the same ss as PW, so that their performances are given in the same column. PW has to be used as the best reference for false positives because it is the prewhitening method with the lowest type 1 error (Zhang and Zwiers, 2004, Yue et al., 2002, Blain, 2013, Wang et al. 2015a), whereas the consideration of the other prewhitening methods as reference allows for the evaluation of the discrepancy in ss among the methods. For the decadal trends, MK, TFPW-Y and VCTFPW have 32-47% of false positives taking PW as reference. This suggests that about two thirds and half of the trends determined using TFPW-Y and VCTFPW, respectively, are false positives. TFPW-WS has less than 2% of false positives, so that it can be considered to have equivalent performance as PW. For the trends on short periods, the lower amounts of false positive for MK and TFPW-Y are due to the overestimation of the slopes with these tests (see section 4.4) leading to trends that are more robust and enhanced ss. The unbiased estimate of the VCTFPW slope produces similar amounts of errors for the short-term trends as for the decadal trends. The percentage of false positives is similar if TFPW-WS is considered as the reference. If MK or TFPW-Y is taken as reference. PW and TFPW-WS have a very low number of false positive independent of the length of the period, leading to the conclusion that

few cases remain uncertain. Note that 5-10% of cases have different ss at the 95% confidence level if MK or TFPW-Y is used, indicating that estimation of the ss using these two methods can have a slight impact on the results. Finally, all the prewhitening methods have a higher number of false positive if VCTFPW is considered as the reference because the added slope at the end of the VCTFPW procedure is smaller than the initial slope and leads to less detectable trends. Note also that the percentage of false positives of PW and TFPW-WS remains low ( $\leq$  4%). For the time series considered in this study, the following conclusions can be made: 1) PW (and PW-cor) performs very well with a small ( $\leq$  3.5%) number of false positives if other prewhitening methods are considered as reference; 2) TFPW-WS has a very low number of false positives (less than 2% if PW is taken as the reference); 3) VCTFPW exhibits high type 1 errors and should consequently not be used to determine the ss; and 4) The difference in ss between MK and TFPW-Y is related to only 5-10% of the trends."

| Period          | MK        | TFPW-Y    | TFPW-WS   | PW/PW-cor | VCTFPW    |
|-----------------|-----------|-----------|-----------|-----------|-----------|
| ≥ 10y
N=2219 | 32.5      | 37.1      | 1.7       | reference | 47.0      |
|                 | 31.8      | 36.1      | reference | 0.7       | 46.4      |
|                 | reference | 9.4       | 0.2       | 0.3       | 26.4      |
|                 | 5.0       | reference | 0.2       | 0.2       | 24.8      |
|                 | 15.7      | 18.4      | 4.0       | 3.5       | reference |
| < 8y
N=1067  | 16.0      | 14.1      | 0.7       | reference | 36.6      |
|                 | 15.9      | 13.9      | reference | 0.5       | 36.7      |
|                 | reference | 3.0       | 0.1       | 0.0       | 28.1      |
|                 | 5.0       | reference | 0         | 0.0       | 29.7      |
|                 | 8.4       | 8.1       | 1.3       | 1.1       | reference |

- Table 3:

7. Line 225-228: Trend analyses were applied on several periods. For all the data sets, **a 10-year period** is considered first and then further possible **multi-decadal periods up to 60 y** for the radio-sounding time series. For the in-situ aerosol properties, **tests with 4 to 9 y periods** are also computed in order to illustrate the problems of trend analysis on very short time series.

Line 781-782: lag-1 autocorrelation of the observations (ak1data) and number of ss partial autocorrelations for the 10y period (order), number of data in the 10y period (N) and reference.

Comment: I think in this section "3 Experimental", the authors should clarify how to analyze the measure data, in order to support the coming results. The meaning of "*a 10-year period*" or "*multi-decadal periods up to 60 y*" are unclear and obscure.

In table 2, the meaning of "number of ss partial autocorrelations for the 10y period (order), number of data in the 10y period (N) and reference" is unclear either.

- The expression "10 y period" corresponds to an analysis over 10 years of measurements. For most cases the more recent 10 y period is considered and

corresponds to 2009-2018 for all parameters. The exception is the AOD where the 10 y period corresponds to 2006-2015 (no more recent AOD data were available). Sometimes all potential 10 y periods are considered, namely 2009-2018, 2008-2017, 2007-2016, etc.

- The expression "multi-decadal periods" correspond to periods of several decades, e.g. 20 y (1999-2018) or 30 y (1989-2018) and up to 60 y (1959-2018) for the tropopause and zero degree levels.
- Section 3 (Experimental) of the manuscript was modified to clarify this point: "Trend analyses were applied on several periods. For all the data sets, the last 10-year period (e.g. 2009-2018 for the BND aerosol scattering coefficient) is considered first and then further possible multi-decadal periods (e.g. the last 20 y (1999-2018), 30 y (1989-2018)) up to 60 y for the radio-sounding time series."

8. Line 275-276: *CL of MK, PW and TFPW-Y, which remove the lag-1 autocorrelation without compensation for the mean values and the variances... Comment: Does that mean "mean and variances of the slope estimate"?*

No, it means the mean value and the variance of the original time series. This is now explicitly written in the manuscript: "*CL of MK, PW and TFPW-Y, which remove the lag-1 autocorrelation without compensation for the mean values and the variances of the original time series, are smaller than for VCTFPW, PW-cor and TFPW-WS. PW-cor and TFPW-WS have the highest CL.*"

9. Line 278-280: The **ss** often decreases for coarser time granularities occasionally leading to not **ss trends** for some of the prewhitening methods. PW, TFPW-WS and VCTFPW methods become **not ss** at finer time granularities than TFPW-Y and MK due to their lower number of false positives.

Comment: It's hard to identify the relationship between the significance of trend and the time granularity from Fig. 2.

The authors agree that this relation is not that obvious from Fig. 2, where it can only be detected for some variables (e.g. TFPW-WS scattering coefficient at 1 and 3 months time granularity or PW, PW-cor, TFPW-WS and VCTFPW tropopause level at 1 month time granularity). This result is much more visible in Fig. 7, but it is an important result that the authors wish to already mention at this stage.

10. Line 281-282: The discrepancies between prewhitening methods are larger than the discrepancies that occur when different temporal segmentations (months or meteorological seasons) are applied.

Comment: Fig.2 does not support this finding.

- This statement is correct but not described with precision. The authors wanted to emphasize that the differences between the slopes computed from the various prewhitening methods are larger than between the different temporal segmentations for a defined prewhitening method. This is clearly visible in Fig. 2 a), b) and c) where, e.g., the slopes for all three temporal segmentations (different symbols) are very close but where the absolute values of PW and VCTFPW slopes are smaller than for the other prewhitening methods. The manuscript was modified to be more precise: "*The slope discrepancies between prewhitening methods are larger than the discrepancies that occur when different temporal segmentations (months or meteorological seasons) are applied for a defined prewhitening method.*"

11. Line 284-285: the similarity of MK slopes with TFPW slopes.

Line 350-352: Due to the detrending procedure, the absolute values of the TFPW-Y slope are larger than the PW slopes and similar to the MK slope values (Fig. 2), even if a tendency to have larger TFPW-Y than MK slopes are observed.

Line 367: TFPW-Y slopes tend to be larger than MK slopes (Fig. 4b), with larger differences at high ak1 data leads.

Comment: If my understanding is right, the MK and TFPW-Y should yield exactly the same slope of trend. The MK test does not estimate the slope of trend directly. It usually reports the magnitude of trend by the use of Sen's slope. The TFPW-Y also estimates Sen's slope as its first step. It will reinstall this trend to the prewhitened series without any modification. So these two slopes should be equal to each other.

The TFPW-Y method reinstalls the Sen's slope (corresponding to the MK slope) to the detrended dataset after removal of the first-lag autocorrelation. The TFPW-Y slope is then estimated from the prewhitened time series (TFPW-Y data) and is not the same as the original slope. The Mk and TFPW-Y slopes are consequently somewhat different because they are computed from two different time series.

12. Line 285-287: For example, the number of data points in the AOD time series (about 65 per year) induces higher CL for time granularities finer than the measurement frequency (about 10 days).

Line 372-373: Removing the lag-1 autocorrelation increases the variance, but decreases the mean.

Line 391-395: The spread of the slopes of the aerosol number concentration for the oneyear aggregation on Fig. 2c shows that the yearly data still have a ss ak1 data for the longest periods of 20 and 24 years (see similar cases in Fig. 2). For shorter periods (5 to 9 years), the ak1 data decreases rapidly for averaging longer than 10 days and even becomes negative for yearly averages.

Comment: These sentences are difficult to be understood. Please rephrase.

The sentences were rephrased:

 Line 285-287:" For example, the very low number of data points in the AOD time series (about 65 per year) corresponds to an average of one data per 5 days; there is consequently a very high amount of missing values for time granularities finer than this measurement frequency and this induces higher CL for time granularities of 1-3 days than granularity of 10 days."

- Line 372-373: "Removing the lag-1 autocorrelation leads to prewhitened data with a larger variance, but lower mean than the original time series."
- Line 391-395: "For the 10 y period represented on Fig. 6, none of the ak1data values are ss for a one-year time granularity. However, there are cases like the 24 y time series of the aerosol number concentration where ak1data is still ss for the one-year time granularity. In these cases, prewhitening methods have to be applied, which leads to the spread of the slopes for the various prewhitening methods visible on Fig. 2a."

13. Line 294-296: The yearly trend was computed for all periods (from 5y to 24y) at all considered time granularities (1 day to 1 month for the meteorological season temporal segmentation), leading to **40 trends**.

Comment: Please clarify what is the 40 trends?

The number 40 corresponds to trends computed for 8 different periods (5, 6, 7, 8, 9, 10, 20 and 24 years) and 5 time granularities (1, 2, 3, 10 and 30 days), so that 8\*5= 40 trends.

14. Line 323-325: *PW* is used as the reference for false positives because it is the prewhitening method with the lowest type 1 error, while TFPW-Y is the reference for false negatives because it is the most powerful test.

Comment: It's inappropriate to state that the TFPW-Y is the most powerful test. The TFPW-Y tends to report significant trends at the expense of committing high type 1 error. This finding has been verified by many literatures. So we can say the TFPW-Y tends to identify significant trends more frequently than other methods, but we cannot say it is the most powerful test.

- The power of the test is defined (see § 21. Line 102 of new manuscript) as the potential to detect ss trend and correspond to low type-2 error. With this definition applied throughout the manuscript, this sentence is right.

15. Line 338-342: For the time series considered in this study, the following conclusions can be made: 1) PW performs very well with an almost vanishingly small ( $\leq 0.3\%$ ) number of false negatives and the ss of PW-cor is similar to that for PW;3) VCTFPW has a very high type 1 and 2 errors and should consequently not be used to determine the ss; and 4) it is not possible to determine whether MK or TFPW-Y is the most powerful method. Line 786-788: Table 3

Comment: The three conclusions made here do not align with the consensus about the prewhitening method among the community. I suggest to recheck the results.

1) The PW tends to overestimate the lag-one autocorrelation coefficient without trend removal, see Hamed (2009). In addition, the PW reduces a portion of real trend, see Yue and Wang (2002). That's the reason why Yue et al. (2002) suggest to remove trend before whitening. So if the TFPW-Y is the reference for false negatives, the PW is less likely to miss only 0.2% significant trends.

3) As it was stated by the authors, e.g. Line 265-266, Table 1, Figure 4(a). The VCTFPW slopes lies between the TFPW and the PW slope values. So no matter one takes the PW

or the TFPW-Y as the reference, the VCTFPW is less likely to commit the highest error among all the prewhitening methods.

4) For the autocorrelated data, the MK and TFPW-Y are not really powerful method. They only tend to report significant trends more frequently than other PW methods. However, both of them commit high type I error as a price.

I have to say, the above opinions are given by Monte-Carlo simulation results. They may not suitable to every real-world series. This study deals with measured data. So I suggest to recheck your results again.

The authors checked the scripts and recomputed all the results. The results presented in the submitted manuscript are correct and do not contradict the cited references. Here some further comments on the numbered remarks:

- Point 1): As stated in the answer to comment 6, it is not possible to detect false negatives without simulated time series with trends. As defined now in Fig. 1, what was called "false negative" are in fact PW false positive if TFPW-Y is taken as reference. Fig. 1, Table 3 and the related descriptions were modified accordingly.
- Point 3) It is right that the VCTFPW lies between the TFPW-Y and the PW slope values and this is a sign that VCTFPW can be accepted as the best slope estimate. But slope estimate has nothing to do with the determination of the statistical significance, since the MK test is constructed to detect the ss but the slope estimate is performed via the Sen's slope. The potential to commit error does not rely on the value of the slope.
- Point 4): the referee is right and the results of this study do completely agree with this statement. The discussion on the power of the method was discarded since the amount of false negative cannot be estimated with real time series.

**16. Line 345: The slope of the trend is always enhanced by the positive ak1data.**

Comment: I think it should be "the slope estimates of the trend is influenced by the positive lag-one autocorrelation". The autocorrelation increases the difficulty of an accurate slope estimation. But it does not increase or decrease the real slope of the trend.

- The referee is right, this sentence is problematic. The slope of the trend is not modified by the autocorrelation in the time series, but it is the slope estimate performed on the original dataset that is influenced. However, it remains correct that the slope estimate performed on the original dataset is enhanced by positive lag-one autocorrelation. The manuscript was modified: "*The slope estimated on the original data is always enhanced by the positive ak*1data"

17. Line 628-629: Consistent with the literature, the use of MK, TFPW-Y and VCTFPW results in a large amount of false positive results while TFPW-WS results in less than 2% of false positives.

Comment: After recheck your results, e.g. table 3, this conclusion should be revised accordingly.

- The results were checked and this statement is still right. The labels "false negative" was however incorrect since false negative cannot be determined on real

measurement (see answer to comment 6), the real value of the slope staying unknown. The number of false positives depends on the prewhitening method chosen as reference. Table 3 was consequently modified and now included the percentage of false positive with each prewhitening method taken as reference. Since PW is commonly accepted to be the method with the least amount of false positive, it is now given in bold, whereas the prewhitening methods known to have a much higher amount of type-1 errors are displayed in italic.

18. Line 637: The confidence limits are much broader for coarser time granularities and the ss is lower.

Comment: Fig. 8 supports this conclusion but Fig. 10 does not. As the time granularity becomes coarser, the confidence limits are much narrower in Fig. 10.

- These conclusions are supported by Figures 2, 7, 8 and 9 where one of the variable is the time granularity. Figure 10 presents the slope, the confidence limits and the ss as a function of the length of the period considered to compute the trend but the time granularity was not considered. These results were computed for a common time granularity of one day.

19. Comment on Figure 2: it is hard to distinguish the time segmentation.

The authors do agree that the density of information on Fig. 2 requires better clarity about the main results suggested by this figure. The slopes computed from the two temporal segmentations (12 months and 4 meteorological seasons) were removed and, instead, boxplots were inserted allow estimation of the discrepancy between the temporal segmentation into four meteorological seasons (considered as the best use for all the used time series) and the 12 months temporal segmentation or no use of segmentation.

20. Comment on Figure 7: it is not easy to identify different PW methods.

The authors changed the symbols and increased their size. Ss trends are consequently no longer given by bigger symbols, but instead are indicated by the red and black lines describing the 95% and 90% confidence level.

21. Comment on Figure 8 and 10: it is unclear how to analyze the slope of trend as well as the confidence limit within each time segmentation. It should be well explained.

In the revised manuscript, the figure caption of Figs 8 and 10 specifies that the slopes correspond to dots and the CL to vertical lines. I hope that I have well understood the referee's requirement.

22. I suggest to improve the quality of the figures, to make them self-explaining.

First the figure captions were all revised in order to increase the clarity and to homogenize the descriptions. Second used symbols are now all described in legends on the figures. All the figures were also verified and modified:

- Fig. 1: To correspond closely to the published code in github, the statistical significance was symbolized with Pprewhitening method instead of S and the choice of P3PW as equal to the min of the PTFPW-Y and PPW (or p-value(3PW)=max(p-value(TFPW-Y), p-value(PW))) is explicitly given.

- Fig. 2: The size of the symbols for the ss are now specified in the legend. The results for the temporal segmentations of 12 months and 4 meteorological seasons are no more displayed but are replaced by boxplots allowing the comparison with the displayed results without temporal aggregation.

- Fig. 3: the used granularities and periods are now specified in the figure caption and the color scale of Fig. 3b is labelled. The titles precise that all periods, all granularities and meteorological seasons were used for a) panel and that all time series, decadal period, granularities and time segmentations were used for b) panel.

- Fig. 4 : the symbols were added in the legend of panel a.

- Fig. 5: the used granularities and periods are specified in the figure caption. The term "all trends" was replaced by "all periods, all granularities" in the figure title.

Fig. 6: A title was added specifying that 10 y period of all times series were used for this figure. The sizes of the symbols for the ss are now specified in the legend.
Fig. 7 the symbols were modified to allow the distinction between the prewhitening methods and the figure caption specifies that no temporal segmentation was used. Ss trends are no longer displayed with bigger symbols, but the ss at 90s and 95% confidence levels is given by the black and red lines.

- Fig. 8: the use of 10 y period is specified in the title. The figure caption now attributes the slope to dots and the CL to vertical lines. The y-axe label also mentions the confidence limits. The size of the symbols for the ss is now specified in the legend. Vertical lines are also added to separate the results for temporal segment. A title is added above the legend to specify that the colors correspond to various granularities.

- Fig. 9: Legends describing the time segmentations and periods are added to the figure.

- Fig. 10: The figure caption now attributes the slope to dots and the CL to vertical lines. The y-axe label also mentions the confidence limits. The sized of the symbols for the ss are now specified in the legend. Vertical lines are also added to separate the results for temporal segment. A title is added above the legend to specify that the colors correspond to various periods. The title of the figure specifies that a granularity of one day was used.

- Fig. 11: it is now specified in the figure caption that the slope were normalized by the median of the data. The color scales have now clear legends. The titles were modified to mention that all the time series, granularities and temporal segmentations as well as periods of at least 10 y were used for both panels a and b.

**Technical corrections**

Line 109. Zwang and Zwiers(2004) does not given in the reference list.

Line 168. I think the correct citation about the VCTFPW method should be "Wang, W., et al., 2015. Variance correction pre-whitening method for trend detection in auto-correlated data. Journal of Hydrologic Engineering, 04015033. doi:10.1061/(ASCE)HE.1943-5584.0001234."

Line 272: "aerosol absorption coefficient" should be "aerosol scattering coefficient".

Thanks for this very detailed review, the technical corrections were applied.

Fig 2 Caption: "*Scattering coefficient, 10y*" and "Tropopause level, 50y". Should it be "24y" and "60y"?

**References**

Hamed, K.H., 2009. Enhancing the effectiveness of prewhitening in trend analysis of hydrologic data. Journal of Hydrology, 368(1-4): 143-155.

Yue, S., Pilon, P., Phinney, B., Cavadias, G., 2002. The influence of autocorrelation on the ability to detect trend in hydrological series. Hydrological Processes, 16(9): 1807-1829.

Yue, S., Wang, C.Y., 2002. Applicability of prewhitening to eliminate the influence of serial correlation on the Mann-Kendall test. Water Resources Research, 38(6): 1068.

---

## Author Comment (AC3) · 14 Oct 2020

**Responses to Anonymous Referee #1**

The authors thank the anonymous referee for the detailed review of the manuscript, for the meticulous pointing out of inconsistencies between tables and figures, as well as for all their comments and suggestions allowing a clear improvement of the paper.

Responses to specific comments:

This manuscript is well written and is an important contribution for more comprehensive trend analysis of atmospheric composition data. The work is robust with very good analysis and discussions of the different effects on the trend results using various prewhitening methods in addition to MK without prewhitening. It is very well appreciated the clear guidelines for choosing methods and approaches for assessing long term trends.
I will recommend the paper to be published as it is. I have only some small comments/questions which you may consider:

- Line 125. why is negative autocorrelation rare in atmospheric processes? Maybe explain a bit more the reasons and differences between negative and positive autocorrelation and/or give a reference.

   A negative autocorrelation changes the direction of the influence. In atmospheric processes, persistence is responsible for autocorrelation rather than "reaction" or "rebound" mechanisms. Persistence embodies the fact that atmospheric variables tend to change relatively slowly, and when changes occur, the autocorrelation tends to decrease toward zero rather than reach negative values. The latter would be the sign of some kind of rebound mechanisms where atmospheric parameters having particular values, for instance above average, would result in latter values being more likely below average. We cannot think of such examples in atmospheric processes, except for processes strongly correlated with natural cycles such as the circadian cycle. For instance, the difference of solar irradiance to the daily solar irradiance average would obviously exhibit a negative autocorrelation at 12h time lag, but it is just due to the high correlation of the solar irradiance with the solar zenith angle. Negative autocorrelation is a violation of independence but it is generally less worrisome because it appears less frequently than positive autocorrelation and it produces greater precision in the average than an independent series would.

- Line 272. Why is aerosol number concentration behaving different than the other components regarding the effect of granularity, i.e. the ss remain until the one-year aggregation?

   The number concentration exhibits a less pronounced seasonal cycle than the other parameters, because its seasonal cycle has variable response to the temperature. For example, at JFJ during summer, higher temperatures lead to a larger influence of the planetary boundary layer and higher production/transport of

primary aerosol. During winter, the colder temperatures can also lead to increase formation of new particles (secondary particles). The 3-month averaging corresponds approximately to a season, so that the small seasonal cycle is not able to mask the positive autocorrelation.

- Fig. 8 and paragraph 429-436. Here you compare the difference in granularity of monthly and seasonal data. Why use different data (scattering contra absorption)? To illustrate the difference in granularity it would have been more logic to use same dataset?

  The comparison between months and meteorological seasons would have been easiest with the same dataset. The authors however chose two different variables to show that the effect of the time granularities on the variability of the slope and the size of the confidence limits is similar for two different variables. This was an option and we try to give examples from all the time series along the paper to enhance that the results do not only concern a peculiar case of atmospheric parameter. The opposite choice was made for Fig. 10.

- Fig10 and paragraph 493-509. Not sure if I understand how the data selection has been done. Do all the periods contain the whole time series? I.e 10 years contain 3x10years data set if the time series is totally 30 years. I assume you somehow taken into account that the actual trend for the whole period will effect the results. Not homogeneous trend over a 30 year period. But why is it then so few data points for the 4 year trend, I,e N=360 and 120 for monthly and seasonal trends?

  For Fig. 10, only the period ending in 2018 with different lengths (4 years to 30 years) is presented, so that the 10 years correspond to the trend between 2009 and 2018 and contains only one 10 y data. If all potential x years trends were used, the mean of the numerous 4 y trends would potentially mask the increase of the absolute values of the slope and the larger difference between individual time segmentations for shorter period length.
  Since only one period of 4 years is used, the number of data in the time series is N=360 (=4 years*3 months*30 days)  for a time segmentation into four meteorological seasons and whereas monthly trends for the same time series are computed with N=120 (=4 years*1 month*30 days) for monthly trends.
  The figure caption was modified in order to clarify the data selection: "*Figure 10: VCTFPW slopes and CL as a function of various period lengths ending in 2018 for the daily aerosol absorption coefficient for the division of the time series into a) 12 months and b) four meteorological seasons. Colors represent time period lengths and bigger symbols represent ss trends.*"

- The new algorithm applied. Is that made available? The scheme sketched in Figure 1 is not very easy to use for others to apply the method. It is recommended that the authors upload the scripts for others to use and adopt if possible.

The new algorithm in Matlab, Python and R will be published in github and the doi will be given in the revised version of the manuscript. We have to finish to documentation of the code before releasing the doi in the next days. . This will happen soon (in conjunction with paper publication). The following section on code availability was added to the manuscript:" *We provide, in dedicated Github repositories hosted within the "mannkendall" organization (https://github.com/mannkendall), a Matlab (DOI: ; https://github.com/mannkendall/Matlab), Python (DOI: ; https://github.com/mannkendall/Python), and R (DOI: ; https://github.com/mannkendall/R) implementation of the algorithm presented in Sec. XX. In particular, these open-source codes, distributed under the BSD 3-Clause License, allow to compute the MK test and the Sen's slope with various prewhitening methods (3PW (default), PW, TFPW-Y, TFPW-WS and VCTFPW). The time granularity, period and temporal segmentation are chosen by the users during the preparation of the datasets. The level of the confidence limits for the MK test, the lag-1 autocorrelation, and the homogeneity between the temporal aggregation can also be defined by the user. The probability for the statistical significance, the statistical significance at the desired confidence level, the Sen's slope and its confidence limits are returned as results. A set a common tests is used to ensure that both the Python and R implementations are consistent with the (original) Matlab implementation of the code."*